# Learning Shared Representations from Unpaired Data

**Amitai Yacobi**[*†]
Department of Computer Science
Bar-Ilan University
Ramat-Gan, Israel
`amitaiyacobi@gmail.com`

**Nir Ben-Ari**[*]
Department of Computer Science
Bar-Ilan University
Ramat-Gan, Israel
`nirnirba@gmail.com`

**Ronen Talmon**
Electrical and Computer Engineering
Technion
Haifa, Israel
`ronen@ee.technion.ac.il`

**Uri Shaham**
Department of Computer Science
Bar-Ilan University
Ramat-Gan, Israel
`uri.shaham@biu.ac.il`

## Abstract

Learning shared representations is a primary area of multimodal representation learning. The current approaches to achieve a shared embedding space rely heavily on paired samples from each modality, which are significantly harder to obtain than unpaired ones. In this work, we demonstrate that shared representations can be learned almost exclusively from unpaired data. Our arguments are grounded in the spectral embeddings of the random walk matrices constructed independently from each unimodal representation. Empirical results in computer vision and natural language processing domains support its potential, revealing the effectiveness of unpaired data in capturing meaningful cross-modal relations, demonstrating high capabilities in retrieval tasks, generation, arithmetics, zero-shot, and cross-domain classification. This work, to the best of our knowledge, is the first to demonstrate these capabilities almost exclusively from unpaired samples, giving rise to a cross-modal embedding that could be viewed as universal, i.e., independent of the specific modalities of the data. Our project page: `https://shaham-lab.github.io/SUE_page`.

## 1 Introduction

The great success of unimodal models [19, 51, 62, 11, 9, 2] in the last decade, and the increasing demand for multimodal applications, have shifted large research attention into the cross-modal domain. Multimodal models present impressive performance on various cross-modal tasks, including image-text [60, 46, 47], speech-text [22], video-text [80] and medical-image-text [84], to name a few.

A primary task in multimodal representation learning is the learning of shared representations, i.e., representation spaces to which instances from different modalities can be mapped, and in which such multimodal instances can be compared. Currently, models learning shared representations typically rely on vast amounts of paired data for training. For example, CLIP [60] was trained on 400 million pairs of images and their captions (i.e., corresponding texts). This kind of supervision (i.e., pairing) is often costly, difficult, and might be even impossible to obtain in many domains and applications. In medical domains, for instance, obtaining a large number of samples is often challenging, as it typically depends on expert annotations. This supervision becomes both costly and time-intensive,

---

[*]Equal contribution, random order.
[†]Also at Moodify.ai, Kefar Saba, Israel

39th Conference on Neural Information Processing Systems (NeurIPS 2025).

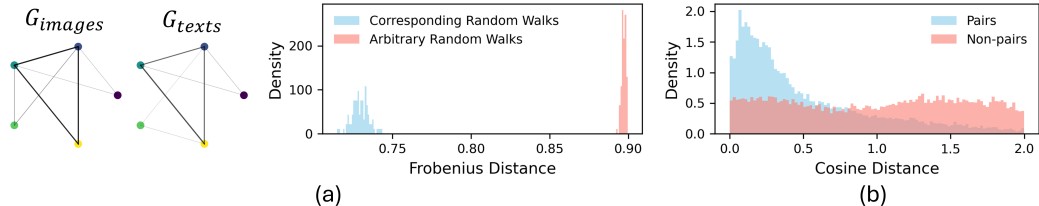

(a)    (b)

Figure 1: **Empirical demonstration of universality.** (a) Distances between corresponding random walks on image and text graphs from MSCOCO, compared to distances to randomly shuffled (non-matching) walks. Although constructed independently from unimodal features, corresponding walks exhibit significantly greater similarity. (b) Distances between paired and unpaired points in the shared space of aligned 2D spectral embeddings (SEs). Paired points are consistently closer, indicating that the independently learned SEs capture analogous structure across modalities (see App. A.7).

or requires rare and specialized tests [32]. Unlike paired data, unpaired data is significantly more accessible and available.

In this paper, we demonstrate that it is possible to learn shared representation spaces while relying almost exclusively on unpaired data. This statement might sound counterintuitive at first, as the supervisory signal from pairing instances from different modalities is crucial for contrastive learning, which is the standard tool used for learning shared representations. However, we argue that this signal can be replaced by a concept we name "universal embeddings". Specifically, we argue that it is possible to leverage unpaired data to learn embedding functions $\Psi$ and $\Phi$ such that if $(x, x')$ is a paired data instance (that is, $x$ and $x'$ correspond to specific different modalities), $\Psi(x) \approx \Phi(x')$. Moreover, we argue that a practical means to learn such maps $\Psi$ and $\Phi$ is to compute the leading eigenfunctions of the diffusion operators corresponding to the unpaired unimodal instances.

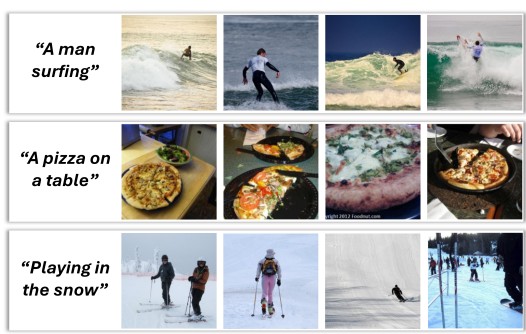

Figure 2: **Almost exclusively unpaired image retrieval.** Retrieved images by SUE for custom captions on MSCOCO, trained with 100 pairs and 10k non-pairs. Despite minimal paired data, the results semantically align closely with text queries.

Specifically, modern pre-trained unimodal foundation models have a proven ability to represent semantics [36]. For example, two given images have close embeddings if their semantic meaning is similar, and far otherwise. These similarities can be captured by a random walk process on the samples' representations, suggesting that a random walk process defined on such unimodal representations should largely correspond to semantic similarity. Therefore, we can expect random walks defined on different unimodal representations that capture semantics well to be highly similar (see Fig. 1). This assumption is supported by substantial empirical evidence from recent works [36, 25].

Random walk processes are finite analogs of diffusion operators. Thereby, the similarity of random walks that are constructed from different, modality-dependent representations implies that the eigenfunctions of the corresponding diffusion operators will have universality properties (i.e., modality-invariance) [13]. Therefore, constructing a spectral embedding (SE) based on the leading eigenvectors of random walks, which are viewed as discrete approximations of the leading eigenfunctions of diffusion operators [5, 69], enables us to take advantage of this concept even in the absence of paired samples. That is, these eigenfunctions can be learned independently from each modality using only unpaired data.

We demonstrate our arguments via a pipeline we call *SUE: Spectral Universal Embedding*, designed to uncover a universal embedding in a weakly-paired scenario, i.e., when large amounts of unpaired data are available alongside only a very small number of paired samples. To the best of our knowledge,

this work is the first to learn shared representations from almost exclusively unpaired data. SUE consists of three steps (see Fig. 3): (i) SE, to extract universal features, (ii) Canonical Correlation Analysis (CCA) on very few samples, to align those features, and (iii) MMD-net, to provide additional non-linear alignment. We demonstrate SUE on vision and language domains and show that with only a small number of paired examples, unpaired data can be leveraged to learn rich multimodal representations. For instance, on the MSCOCO dataset, SUE improves retrieval performance by over 250% on average compared to contrastive learning trained with the same number of pairs. Fig. 2 demonstrates SUE's ability to learn cross-modal properties with almost no paired data.

Our contributions can be summarized as follows: (i) we demonstrate that learning of shared representation spaces can be done based on unpaired data, (ii) we show that spectral embeddings are an effective means to obtain universality, and (iii) we establish a pipeline for shared representation learning, utilizing primarily unpaired data, along with a very small number of paired instances.

## 2  Related Work

**Paired-trained models.**  Multimodal models have recently seen great improvements in rich-data modalities, such as text with image [60, 46, 47], speech [22] and video [80]. Current state-of-the-art multimodal models rely on contrastive learning between corresponding pairs of unimodal representations. While these methods yield high-quality results, they require extremely large amounts of paired data, limiting their applicability to arbitrary domains. Alternative methods, such as CSA [48], slightly reduce the number of pairs required but still operate exclusively on paired data. Unlike these, the universal embedding offers an integration of any two domains, uncovering a shared space primarily from unpaired samples, complemented by only a small number of pairs. Moreover, following recent research on the modality gap in multimodal learning [49, 23, 65], contrastive-based methods fail to be universal due to modalities residing on different manifolds. In contrast, a universal embedding should not exhibit this gap (see App. B for details).

**Incorporating Unpaired Data.**  Unpaired samples have been used to replace certain applications of multimodal models without constructing shared representations, including medical image segmentation [82, 75], image-to-speech translation [52], image-to-image translation [87, 88, 12], bilingual word embeddings [1], and domain confusion [26, 27, 72, 85]. Other methods incorporate unpaired data to complement additional signals, such as large paired datasets [56, 71, 18, 28, 78, 83] or classification labels [74, 79, 45]. In contrast, SUE uncovers a latent universal space between two unimodal domains that share semantic content, using unpaired data and only a small number of pairs.

Several works explore shared representation learning from limited to no supervision. Hoshen and Wolf [34] proposed an unpaired variant of CCA, but as noted in their work, the method is unstable and requires many runs to yield satisfactory results, limiting its practicality. MACK [35], while targeting image-text alignment, depends on a segmentation-to-text model trained with paired data, limiting its scope. Closer to our direction are [10, 54], which align modalities by referencing a set of paired examples. However, they overlook local structure in the unimodal embedding space - an important cue, since non-immediate neighbors are often uninformative in high-dimensional settings. In contrast, SUE leverages local unimodal neighborhoods and a few weak pairs to construct a unified cross-modal representation.

**Spectral Embedding (SE) in multimodal learning.**  SE is known for its global structure preservation properties of unimodal manifolds [15, 55, 73]. One perspective, for instance, is that the SE is the space in which the Euclidean distances represent the diffusion distances on the original manifold [14]. The diffusion distance is a manifold-dependent metric, offering advantages over traditional metrics (e.g., Euclidean, cosine), especially in high-dimensional spaces [15]. Recent works investigated the global structure preservation abilities of SE in the cross-modal scenario [24, 44, 81]. Specifically, they showed that joint SE extracts meaningful representations from each modality. The universal embedding is inspired by these findings, and the analysis we provide extends this idea. Particularly, we show that independent SEs capture similar semantic properties *across different modalities*, such as images and texts.

# 3 Preliminaries

**Spectral Embeddings.** SE [4, 14] is a popular method, used across various domains [53, 16, 86, 3, 21, 42, 43, 68, 8, 89]. In particular, let $\mathcal{X} \subseteq \mathbb{R}^d$ denote a collection of data points sampled from a manifold $\mathcal{M}$. Let $W \in \mathbb{R}^{n \times n}$ be a positive symmetric graph affinity matrix, with nodes corresponding to $\mathcal{X}$, and let $D$ be the corresponding diagonal degree matrix (i.e., $D_{ii} = \sum_{j=1}^{n} W_{ij}$). The random walk matrix is defined as $P = D^{-1}W$, and the random walk graph Laplacian as $L = I - P$, where $I$ is the identity matrix. The eigenvalues of $P$ can be sorted to satisfy $1 = \lambda_1 \geq \lambda_2 \geq \cdots \geq \lambda_n$ with corresponding eigenvectors $v_1, v_2, \ldots, v_n$ [76].

For a given target dimension $k$, the leading non-trivial $k$ eigenvectors provide a non-linear low-dimensional embedding of the random walk process, known as *Spectral Embedding* (SE). Specifically, the SE representation of a sample $x_i \in \mathcal{X}$ is given by $y_i = \big(v_1(i), \ldots, v_k(i)\big)$.

**Laplace-Beltrami and Diffusion Operators.** Given a Riemannian manifold $\mathcal{M}$ the Laplace-Beltrami operator $\Delta$ acts on functions $f : \mathcal{M} \to \mathbb{R}^k$ by $\Delta f = \mathrm{div}(\nabla f)$, where div is the divergence and $\nabla$ the gradient. The diffusion operator (heat kernel) at time $t$ is then defined by $P_t = e^{-t\Delta}$. For more details please see [14]. Importantly to this work, the Laplacian and random walk matrices converge to the Laplace-Beltrami and diffusion operators, respectively [14, 5, 69].

**Parametric SE.** Recent works proposed parametric SE implementations, such as SpectralNet [67, 6]. These are deep-learning methods, designed to tackle the scalability and generalizability drawbacks of SE. Specifically, they learn a parametric map $f : \mathbb{R}^d \to \mathbb{R}^k$, which minimizes the Rayleigh quotient

$$\mathcal{L}_{\text{spectralnet}}(f) = \frac{1}{n^2} \operatorname{Trace}\big(f(X)^T L f(X)\big)$$

while enforcing orthogonality (i.e., $f(X)^T f(X) = I_k$), where $L$ is the random walk Laplacian corresponding to the data matrix $X \in \mathbb{R}^{n \times d}$.

**Maximum Mean Discrepancy (MMD).** MMD [29, 30] is a distance measure between two probability distributions $p, q$. It is defined with respect to a function class $\mathcal{F}$ by

$$\mathrm{MMD}(\mathcal{F}, p, q) = \sup_{f \in \mathcal{F}} \big(\mathbb{E}_{x \sim p} f(x) - \mathbb{E}_{x \sim q} f(x)\big)$$

When $\mathcal{F}$ is a reproducing kernel Hilbert space with kernel $k$ (e.g., RBF kernel), the squared MMD can be written as

$$\mathrm{MMD}^2(\mathcal{F}, p, q) = \mathbb{E}_{x,x' \sim p} k(x, x') - 2\mathbb{E}_{x \sim p, y \sim q} k(x, y) + \mathbb{E}_{y,y' \sim q} k(y, y')$$

where $x$ and $x'$ are independent, and so are $y$ and $y'$. We use an empirical variant of the squared MMD (see Sec. 4.2).

# 4 Spectral Universal Embedding

## 4.1 Rationale

Here we detail the mathematical motivation for each component of SUE and describe an overview of a pipeline to reveal a universal embedding out of given modalities (see Fig. 3 for an overview).

**Mathematical Motivation.** We formalize our assumption as follows. Let $\mathcal{M}$ be a latent, underlying semantic manifold, and let $f, g$ be two transformations, such that $f(\mathcal{M})$ and $g(\mathcal{M})$ represent the two modalities from which we observe samples. There is a body of work specifying conditions under which the spectral properties of $\mathcal{M}$ are preserved under $f, g$. For example, if $f, g$ have bounded distortion and bounded Ricci curvature, the corresponding eigenfunctions of the Laplace-Beltrami operator, as well as of the diffusion operator, on $f(\mathcal{M})$ and $g(\mathcal{M})$ are similar in the $L_\infty$ sense [7] (see App. D).

Intuitively, our assumption states that the diffusion operators defined on each modality are relatively similar. This assumption is empirically supported by the results in Sec. 5, as well as in recent

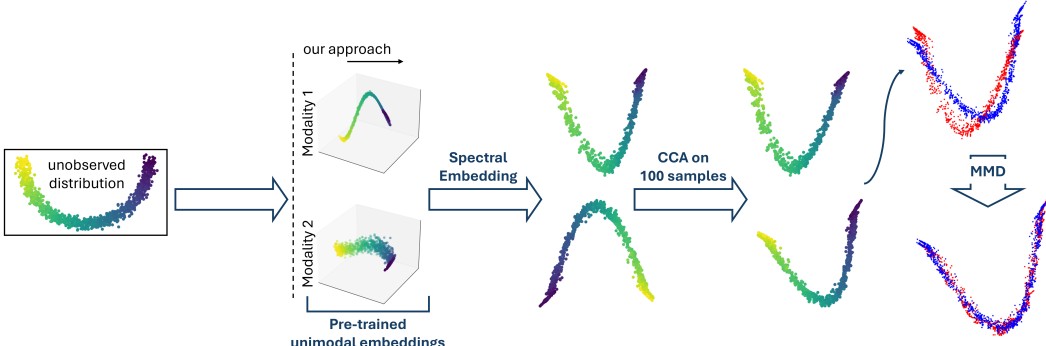

Figure 3: **SUE's overview**. The modalities (represented by their unimodal embeddings) represent an unobserved universal (semantic) distribution; the SE is capable of retrieving this universal structure, up to rotations; CCA on a minimal number of pairs enable linear alignment between the modalities, but not sufficient for a joint universal embedding; the MMD then fixes the misalignment between the modalities, integrating them into the universal embedding space.

works [36, 25]. Then, universality is enabled through the eigenfunction preservation properties of the similar diffusion operators. Namely, the eigenfunctions of these operators will be universal, in the sense of modality-invariance (see Sec. 1). In practice, the ability to learn the diffusion operators' eigenfunctions is obtained via SpectralNet [67]: while trained to compute the eigenvectors of the random walk matrix of its training data, being a generalizable parametric map makes it a practical means to compute the eigenfunctions of the diffusion operator, viewing the eigenvectors as a discretization of the eigenfunctions [5, 69]. Crucially, we train SpectralNet on unimodal data only; hence, no paired data is needed to learn the Laplacian eigenfunctions, i.e., our universal embedding functions.

**Overview.** SUE consists of three steps: SE, CCA and MMD. First, it maps each pre-trained unimodal embedding space into its corresponding eigenspace, to retrieve the global structure of each modality [4, 55, 70]. Using SpectralNet [67], this is done parametrically, allowing generalization to test data. Noteworthy, SE is not unique, as eigenvalues with multiplicity $p$ can yield any basis spanning the $p$-dimensional eigenspace and even single eigenvectors may differ by sign.

To resolve the SE ambiguity, and provide additional linear alignment, we use CCA on a minimal number of paired samples. However, as the CCA purposefully considers a limited number of samples, and the SEs differ by more than an orthogonal transformation, we strengthen the cross-modal alignment using a Maximum Mean Discrepancy (MMD) residual network [66]. This kind of network architecture was originally designed for batch-effect removal, by minimizing the empirical MMD value of two distributions. Namely, we view the two low-dimensional representations as similar distributions and learn a (close to identity) non-linear shift to align the distributions. The MMD serves as the last step to fine-tune the alignment. Notably, MMD loss does not require paired data, which enables the utilization of the full unpaired dataset.

## 4.2 Computing SUE

In this section, we describe the computation of SUE, roughly described in Sec 4.1, in more detail. A summary of the steps of the SUE algorithm is outlined in Algorithm 1.

**Notations.** Throughout this section, we will use the following notations. Let $\mathcal{X} \subseteq \mathbb{R}^{d_1}, \mathcal{Y} \subseteq \mathbb{R}^{d_2}$ be sets of unpaired pre-trained unimodal embeddings of sizes $n_1, n_2$, resp. Accordingly, denote $\mathcal{X}_p = \{x_1, \ldots, x_m\} \subseteq \mathcal{X}, \mathcal{Y}_p = \{y_1, \ldots, y_m\} \subseteq \mathcal{Y}$ to be sets of paired embeddings. Importantly, $m \ll n_1, n_2$. Let $k \geq r$ be two pre-chosen dimensions for the SE and final universal representations.

**Approach.** Given $\mathcal{X}, \mathcal{Y}$, we train two independent SpectralNet models $S_{\mathcal{X}} : \mathcal{X} \to \mathbb{R}^k, S_{\mathcal{Y}} : \mathcal{Y} \to \mathbb{R}^k$ to approximate the $k$-dimensional SE of each modality. Due to the non-uniqueness of the SE, $S_{\mathcal{X}}$ and $S_{\mathcal{Y}}$ might differ by sign and basis of each eigenspace.

To address this ambiguity we utilize $\mathcal{X}_p$ and $\mathcal{Y}_p$. Specifically, we employ CCA on $\left(S_\mathcal{X}(\mathcal{X}_p), S_\mathcal{Y}(\mathcal{Y}_p)\right)$ to obtain the projections $Q_\mathcal{X}, Q_\mathcal{Y} \in \mathbb{R}^{k \times r}$. These projections are used to align $S_\mathcal{X}(\mathcal{X})$ and $S_\mathcal{Y}(\mathcal{Y})$. The linearly aligned SEs approximations can be written as $\tilde{S}_\mathcal{X} := Q_\mathcal{X} \circ S_\mathcal{X}, \ \tilde{S}_\mathcal{Y} := Q_\mathcal{Y} \circ S_\mathcal{Y}$.

Then, we learn a residual neural network $F_\theta : \mathbb{R}^r \to \mathbb{R}^r$ to bring the distribution of the linearly aligned SEs as close as possible. Specifically, we minimize the squared MMD between the two empirical distributions

$$\mathcal{L}_{\text{MMD}} = \frac{1}{m_1^2} \sum_{x_i, x_j \in \mathcal{X}} \kappa(\tilde{x}_i, \tilde{x}_j) - \frac{1}{m_1 m_2} \sum_{x_i \in \mathcal{X}, y_j \in \mathcal{Y}} \kappa(\tilde{x}_i, \tilde{y}_j) + \frac{1}{m_2^2} \sum_{y_i, y_j \in \mathcal{Y}} \kappa(\tilde{y}_i, \tilde{y}_j) \quad (1)$$

where $m_1, m_2$ are the corresponding batch sizes, $\kappa$ is a universal kernel (e.g., RBF kernel), and $\tilde{x}_i = \tilde{S}_\mathcal{X}(x_i), \tilde{y}_i = \tilde{S}_\mathcal{Y}(y_i)$. The final functions can be written as $f_\mathcal{X} := \tilde{S}_\mathcal{X}, \ f_\mathcal{Y} := F_\theta \circ \tilde{S}_\mathcal{Y}$.

Given a new test point $y_t$, sampled from the same distribution as $\mathcal{Y}$, we simply propagate it through $f_\mathcal{Y}$, and similarly to a test point sampled from the $\mathcal{X}$ distribution.

---

**Algorithm 1:** Spectral Universal Embedding (SUE)

---

**Input:** Unpaired sets of pre-trained unimodal embeddings $\mathcal{X} \in \mathbb{R}^{n_1 \times d_1}$ and $\mathcal{Y} \in \mathbb{R}^{n_2 \times d_2}$, and paired sets $\mathcal{X}_p$ and $\mathcal{Y}_p$ of size $m \geq 0$
**Output:** Maps $f_\mathcal{X} : \mathbb{R}^{d_1} \to \mathbb{R}^r, \quad f_\mathcal{Y} : \mathbb{R}^{d_2} \to \mathbb{R}^r$ approximating the universal embedding from each modality
1 Train $S_\mathcal{X}, S_\mathcal{Y}$
2 Perform CCA on $\left(S_\mathcal{X}(\mathcal{X}_p), S_\mathcal{Y}(\mathcal{Y}_p)\right)$ to obtain projections $Q_\mathcal{X}, Q_\mathcal{Y} \in \mathbb{R}^{k \times r}$
3 Train a residual neural network $F_\theta : \mathbb{R}^r \to \mathbb{R}^r$ to minimize the MMD loss $\mathcal{L}_{\text{MMD}}$ (Eq. 1)
4 Return the maps:
$$f_\mathcal{X} := Q_\mathcal{X} \circ S_\mathcal{X}, \quad f_\mathcal{Y} := F_\theta \circ Q_\mathcal{Y} \circ S_\mathcal{Y}$$
5 At inference time, propagate the sample $x$ or $y$ through the appropriate map $f_\mathcal{X}(x)$ or $f_\mathcal{Y}(y)$

---

## 5 Experiments

In this section, we provide empirical evidence for the effectiveness of unpaired data for constructing shared representation by evaluating SUE's ability to uncover it. We assess SUE's effectiveness in cross-modal settings using both quantitative and qualitative analyses. Our experiments cover a diverse set of tasks, including retrieval, generation, semantic arithmetic, zero-shot learning, and cross-domain classification. Notably, this is the first work, to the best of our knowledge, to demonstrate such versatility across tasks and modalities while relying almost exclusively on unpaired data.

The remainder of this section is organized as follows. Sec. 5.1 evaluates retrieval performance. Sec. 5.2 presents a range of applications of universality. To better understand the contribution of each component in SUE, we investigate the effects of various components. In Sec. 5.3, we examine the impact of incorporating additional paired and unpaired samples, revealing that unpaired data holds greater potential than previously recognized. Sec. 5.4 presents an ablation study assessing the roles of SUE's components (SE, CCA, and MMD), demonstrating that the novel integration of these techniques is central to SUE's effectiveness. Due to space constraints, we refer to App. A for additional results. Further details on the experimental setup, implementation, and training procedures are provided in Appendix E.

**Datasets.** To evaluate SUE's performance, we use several paired datasets. To provide informative qualitative results and facilitate an intuitive understanding of the universal embedding concept, we use three vision-language datasets (🟩🟦): Flickr30k [59], MSCOCO [50], and Polyvore [31]; as well as a vision-vision dataset (🟦🟦): Edges2Shoes [37]; and a tabular-to-tabular dataset (⬜🟦): Handwritten [20]. For the generation task, we use another vision-language dataset (🟩🟦): caption-FFHQ [40]. See App. E.1 for further details on the datasets and preprocessing (e.g., the unimodal models used and adaptation to weakly-paired settings).

**Baselines.** To the best of our knowledge, this is one of the very first works to exploit weakly-paired data (i.e., very few pairs). As such, no direct baselines exist for this setting. To contextualize results,

Table 1: **Retrieval results.** Results with few paired samples on vision-language (🟩🟦), vision-vision (🟩🟩), and tabular-tabular (🟦🟦) datasets from each modality to another: image-to-text (I2T), text-to-image (T2I), edges-to-shoes (E2S), shoes-to-edges (S2E), KL coefficients-to-pixel averages (K2P), pixel-to-KL (P2K); by SUE, Contrastive, and CSA. The Imp. column states the relative mean improvement of SUE over Contrastive learning. Using the same small number of pairs, SUE significantly outperforms the popular paired method. **SUE substantially relies on unpaired data.**

| | #paired | | SUE (ours) | | | Contrastive | | | CSA | | | Imp. |
|---|---|---|---|---|---|---|---|---|---|---|---|---|
| | | | R@1 | R@5 | R@10 | R@1 | R@5 | R@10 | R@1 | R@5 | R@10 | |
| MSCOCO 🟩🟦 | 100 | I2T | 5.75 | 21.50 | 34.25 | 1.50 | 8.50 | 13.00 | 0.00 | 1.25 | 3.00 | **+257.20%** |
| | | T2I | 5.25 | 18.25 | 33.25 | 0.80 | 5.80 | 12.20 | 0.00 | 1.00 | 2.25 | |
| Flickr30k 🟩🟦 | 500 | I2T | 4.25 | 19.75 | 32.00 | 3.00 | 9.50 | 16.20 | 0.25 | 1.25 | 2.50 | **+103.32%** |
| | | T2I | 5.75 | 22.00 | 32.75 | 2.50 | 9.80 | 15.00 | 0.25 | 0.75 | 2.75 | |
| Polyvore 🟩🟦 | 500 | I2T | 6.00 | 22.75 | 32.25 | 3.20 | 13.8 | 22.5 | 0.25 | 1.25 | 2.25 | **+55.67%** |
| | | T2I | 4.75 | 20.75 | 32.00 | 4.00 | 11.50 | 23.00 | 0.25 | 1.00 | 3.25 | |
| Edges2Shoes 🟩🟩 | 50 | E2S | 4.00 | 16.00 | 25.25 | 1.00 | 5.50 | 14.00 | 0.25 | 1.50 | 2.75 | **+200.51%** |
| | | S2E | 3.50 | 17.00 | 27.00 | 0.80 | 6.00 | 12.80 | 0.25 | 1.50 | 3.00 | |
| Handwritten 🟦🟦 | 100 | K2P | 25.50 | 62.00 | 79.00 | 4.80 | 17.00 | 28.00 | 4.25 | 12.25 | 17.50 | **+283.60%** |
| | | P2K | 25.00 | 61.75 | 78.00 | 4.80 | 17.80 | 30.50 | 3.50 | 9.00 | 15.00 | |

we compare SUE with the contrastive method designed for fully paired data [60, 80, 22] (training an MLP on pre-trained unimodal features using the same number of pairs), and the CSA method [48] designed for small paired datasets. In addition, we examine domain confusion (App. A.1) and SAIL [83] (App. A.2), a representative approach for incorporating unpaired data alongside paired data; both perform notably worse in these scenarios.

## 5.1 Retrieval

**Retrieval.** Tab. 1 reports retrieval scores for SUE and contrastive learning on several datasets using a randomly sampled test set of 400 examples. Following prior work [60, 22], we use the Recall@$k$ metric. The results demonstrate SUE's ability to capture cross-modal semantics with as few as 100 pairs for MSCOCO, 500 for Flickr30k and Polyvore, and just 50 for Edges2Shoes. Remarkably, these results are also comparable with the ones reported by Huang et al. [35] on the Flickr30k T2I task[3], despite MACK using a paired segmentation-to-text model (see App. A). In App. A.5 we report mean average precision (mAP) results, reinforcing the same conclusions. In App. A.6, we further analyze how different unimodal encoders affect SUE's retrieval performance.

These retrieval results introduce a significant advance in the ability to learn meaningful representations in the practical scenario of limited pairs. Specifically, when comparing SUE, trained on the MSCOCO dataset with only 100 pairs (and additional unpaired data), to a network trained with CLIP loss solely on the same 100 pairs, SUE achieves better performance by more than 250%. This highlights that SUE derives its strength from the unpaired samples.

Fig. 5a analyzes the power of the contrastive method with a small number of pairs, compared to SUE. Remarkably, to get the results SUE achieves with 100 pairs (and 10k non-pairs), the contrastive method requires an order of magnitude more pairs. This shows the utility of SUE in limited pairs scenarios.

In Fig. 4a we supplement the quantitative results with several qualitative examples. Additional qualitative results can be found in App. A.9. These results illustrate how SUE successfully captures semantic relationships across modalities, even in cases where the exact pair is not among the first neighbors (leading to a zero Recall@$k$ score).

In Fig. 2, we consider a custom caption scenario, where we retrieve images of a more general query sentence that cannot be found in the dataset. These examples further support SUE's ability to capture these cross-modal semantics. These qualitative examples also depict the challenge of reliably measuring semantic coherence across modalities in retrieval tasks using the recall@$k$ metric. Therefore, in App. C we consider a new soft variant of Recall@$k$, which follows previous works [33] and uses CLIP score as a measure of semantic alignment between image-text pairs.

---

[3]R@1 10.3; R@5 25.1; R@10 34.0.

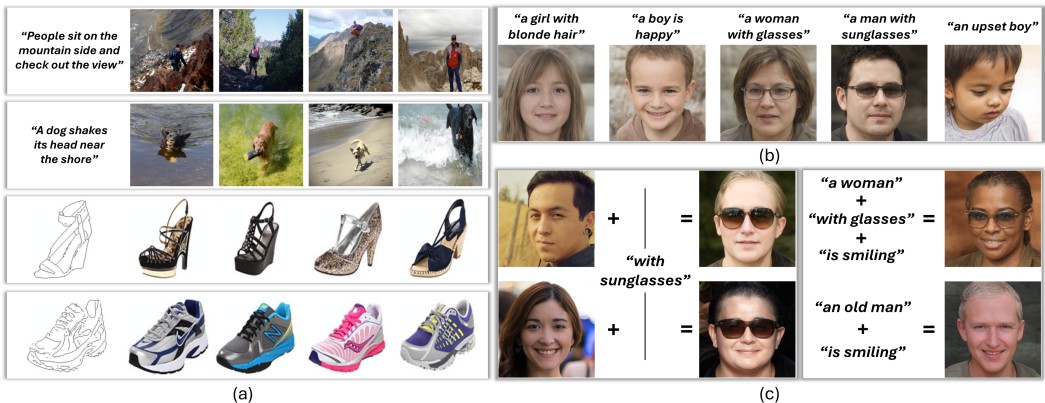

Figure 4: **(a) Images retrieval examples. SUE captures cross-modal semantic structure.** Top four retrieved images for text and shoe-edge queries from Flickr30k and edges2shoes. True pairs are excluded, yet results remain semantically aligned. **(b) (Almost-) Text-free text-to-image generation examples.** Images generated from various text queries using a generator and a converter trained exclusively on images. **(c) (Almost-) Text-free arithmetics examples.** Images generated from sums of text and image queries - for example, "with sunglasses" + man/woman image yields a corresponding result with sunglasses.

These examples show that despite minimal paired training data, SUE effectively reveals meaningful cross-modal semantic connections, suggesting its utility in real-world scenarios where large paired datasets are unavailable or impractical to obtain.

## 5.2 Applications of Universality

**(Almost-) Text-free text-to-image generation.** Fig. 4b depicts several examples of text-to-image generation with minimal text-image correspondence. In particular, we use a pre-trained GAN inversion model [77] trained on the FFHQ dataset [40] and train a converter model that maps from the universal image embeddings to the GAN's latent space. During inference, we pass universal *text* embeddings to the converter and generate face images from its output using the GAN decoder. Noteworthy, both the GAN and the converter were trained exclusively on images, with few text-image pairs available only during SUE's training. For more details on the training process of the converter model, please refer to App. E.3.

**(Almost-) Text-free arithmetics.** Fig. 4c presents the results of emergent arithmetic operations on the universal embedding. Notably, these examples require no additional training - we simply perform vector summation of text and image embeddings and generate the image using the pre-trained converter and GAN. This approach aligns with techniques demonstrated using CLIP [61] but with limited text-image pairing. This summation in the vector space translates into an intuitive integration of semantics, making it, in a sense, a semantic vector space, while almost no text-image pairs were available during the construction of the shared embedding. This showcases the SUE's cross-modality capturing abilities, even in the absence of large paired datasets.

**Zero-shot.** In App A.3 we further demonstrate SUE's ability to capture high-level cross-modal, using a zero-shot scenario - acting as a classifier without being explicitly trained for classification.

**Emergent cross-modal classification.** In App A.4 we demonstrate the effectiveness of universality for the classification of an unlabeled domain with minimal correspondence to a labeled domain.

## 5.3 Effect of Paired and Unpaired Samples

**Unpaired samples.** Fig. 5b shows the impact of additional unpaired samples. This experiment is of significant interest, as unpaired samples are usually considered unusable in the multimodal setting for point-to-point matching. The results indicate that additional *unpaired* data significantly enhances

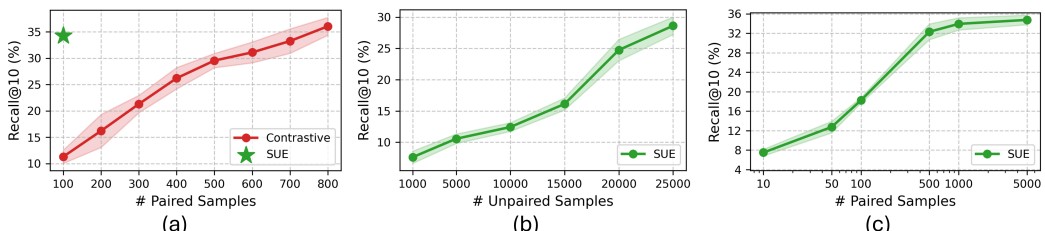

Figure 5: **(a) Contrastive requires an order of magnitude more pairs to achieve similar results as SUE in the weakly-paired regime.** Recall@10 results on MSCOCO by SUE (with 100 pairs) and Contrastive with various numbers of pairs. SUE exploits unpaired data to outperform contrastive learning when limited pairs are available. An order of magnitude more pairs are required to achieve similar results with contrastive learning; (b-c) Effect of #unpaired and #paired samples on Recall@10 results on image retrieval on the Flickr30k dataset. **(b) SUE improves as the amount of unpaired data is increased. (c) SUE relies on non-pairs instead of pairs.** SUE relies minimally on paired data, while substantially on unpaired data, enabling it to enhance its performance with additional unpaired samples, which are much easier to obtain.

retrieval results. This opens the door for a new regime of multimodal learning - using unpaired data with only a minimal number of available pairs.

**Paired samples.** Fig. 5c depicts the results of an analogous experiment examining the effect of the number of paired samples required for the CCA step, with the unpaired samples held constant. As expected, a minimal number of paired samples are required for good results ($\sim$500 in this case of Flickr30k). However, SUE does not rely on additional pairs, as increasing their number above the minimum required is redundant. This outcome highlights the potential for learning significant cross-modal embeddings while focusing on unpaired data, which is much easier to obtain.

## 5.4 Ablation Study

Tab. 2 presents the results of various ablations of SUE, from the original unimodal representations to the complete SUE pipeline. Alongside these empirical results, we provide UMAP visualizations of SUE's three steps in Fig. 13 in App. A.11. The main outcome from these results is that SE has a pivotal role in revealing the universal embedding. Attempts to align the original unimodal representations yield mostly trivial retrieval results (see left column of each dataset in Tab. 2). However, applying the same alignment approach (i.e., CCA alongside MMD) on the SE space, significantly improves the retrieval performance. This underscores SE's essential role in extracting the modality-invariant representations. This is further supported by the results in Tab. 10 in App. A.10, where replacing the SE step with an Auto-Encoder significantly decreases the performance. These results support the existence of the universal embedding. SUE successfully learns a meaningful cross-modal semantic representation with only a minimal number of paired samples, used solely to facilitate linear cross-modal alignment. See extended ablation study in App. A.10.

Table 2: **Ablation study results. SE is pivotal.** Text-to-image R@10 results on the Flickr30k and MSCOCO datasets. The highlighted numbers are SUE's full pipeline. Notably, SE is necessary for the success of CCA (and CCA with MMD).

| | Flickr30k | +SE | MSCOCO | +SE |
|---|---|---|---|---|
| | 2.25 | 8.75 | 1.50 | 4.25 |
| +MMD | 3.75 | 5.50 | 2.00 | 3.75 |
| +CCA | 4.50 | 30.25 | 7.75 | 31.50 |
| +CCA +MMD | 4.75 | **32.75** | 9.75 | **33.25** |

# 6 Conclusion and Future Work

This paper demonstrates the ability to learn shared representations almost exclusively from unpaired data. Specifically, we showed that a shared structure can be captured through a random walk process on each modality-specific representation, with Spectral Embedding (SE) as the practical foundation for uncovering it, revealing its universal properties.

In doing so, we challenge the prevailing approach in multimodal learning, which relies heavily on extremely large amounts of paired samples. SUE, operating with weakly-paired data, in the sense of very few pairs, yields meaningful results across various applications, especially highlighted by the comparison to contrastive learning.

**Limitations.** Despite these contributions, several limitations remain. While SUE provides compelling evidence for the hidden potential of unpaired data in learning a shared representation space, it does not yet match the performance of state-of-the-art models trained with massive paired datasets. Furthermore, our evaluation has been limited to a small set of modalities and tasks, leaving open questions about its generalizability to more complex domains such as video, speech, or scientific data. Addressing these challenges offers a promising avenue for advancing the framework.

Importantly, our work does not yet aim to replace existing multimodal models trained with large sets of paired data. Rather, it introduces a new concept for settings where such supervision is unavailable. Interestingly, the observation in App. A.6, along with empirical results from Huh et al. [36], Fan et al. [25], suggests that increasing the capacity of unimodal models improves the alignment between unimodal representations, leading to better SUE performance without requiring additional paired data. Finally, this work establishes a foundation for a fully unpaired multimodal learning framework across arbitrary modalities, which we consider an intriguing and promising avenue for future work.

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

# A  Additional Results

**Code availability.**  Our code, including SUE implementation and all experiments, is available at `https://github.com/shaham-lab/SUE`.

**Note about MACK.**  While the Flickr30k T2I task results are reported in the paper by Huang et al. [35], the absence of public implementation prevents a more comprehensive comparison with MACK.

## A.1  Domain Confusion results

Tab. 3 shows the results of MMD (as a representative of the domain confusion methods) and SUE on all retrieval tasks considered in Tab. 1. MMD does not necessitate any paired data for training. However, the results show that it significantly underperforms SUE, which uses only very few pairs.

Table 3: **MMD Retrieval results.** Retrieval results on vision-language (🟩🟦) and vision-vision (🟩🟩) datasets from each modality to another: image-to-text (I2T), text-to-image (T2I), edges-to-shoes (E2S), shoes-to-edges (S2E), note-to-measurement (N2M), measurement-to-note (M2N). MMD significantly underperforms SUE.

| | | SUE (ours) | | | MMD | | |
|---|---|---|---|---|---|---|---|
| | | R@1 | R@5 | R@10 | R@1 | R@5 | R@10 |
| Flickr30k 🟩🟦 | I2T | 4.25 | 19.75 | 32.00 | 0.50 | 1.25 | 4.00 |
| | T2I | 5.75 | 22.00 | 32.75 | 0.50 | 1.75 | 3.75 |
| MSCOCO 🟩🟦 | I2T | 5.75 | 21.50 | 34.25 | 0.00 | 1.50 | 3.00 |
| | T2I | 5.25 | 18.25 | 33.25 | 0.50 | 1.75 | 2.00 |
| Polyvore 🟩🟦 | I2T | 6.00 | 22.75 | 32.25 | 0.00 | 1.25 | 2.50 |
| | T2I | 4.75 | 20.75 | 32.00 | 0.00 | 1.25 | 2.50 |
| Edges2Shoes 🟩🟩 | E2S | 3.75 | 18.00 | 26.75 | 0.50 | 1.25 | 4.00 |
| | S2E | 2.75 | 15.25 | 25.50 | 0.50 | 1.75 | 3.75 |

## A.2  Comparison with SAIL

Tab. 4 presents a comparison between SUE and SAIL [83], a representative method for incorporating large unpaired data alongside paired data. The comparison is on vision-language datasets (🟩🟦) under the same settings as Tab. 1. The results demonstrate that SUE is more effective in the weakly paired scenario.

Table 4: **SAIL comparison results.** Results with few paired samples on vision-language (🟩🟦) from each modality to another: image-to-text (I2T), text-to-image (T2I); by SUE and SAIL. Using the same small number of pairs, **SUE significantly outperforms SAIL.**

| | #paired | | SUE (ours) | | | SAIL | | |
|---|---|---|---|---|---|---|---|---|
| | | | R@1 | R@5 | R@10 | R@1 | R@5 | R@10 |
| MSCOCO 🟩🟦 | 100 | I2T | 5.75 | 21.50 | 34.25 | 0.75 | 4.75 | 7.25 |
| | | T2I | 5.25 | 18.25 | 33.25 | 0.75 | 4.50 | 7.00 |
| Flickr30k 🟩🟦 | 500 | I2T | 4.25 | 19.75 | 32.00 | 1.00 | 3.75 | 6.50 |
| | | T2I | 5.75 | 22.00 | 32.75 | 0.50 | 3.00 | 6.75 |
| Polyvore 🟩🟦 | 500 | I2T | 6.00 | 22.75 | 32.25 | 0.50 | 4.25 | 7.25 |
| | | T2I | 4.75 | 20.75 | 32.00 | 0.75 | 4.00 | 7.50 |

## A.3  Zero-shot

To demonstrate SUE's ability to capture high-level cross-modal semantics, we consider the zero-shot scenario [60]. This is a by-product of SUE, with no further adjustment. We use the zero-shot scenario

to demonstrate the universality properties of SUE. Specifically, we train SUE on Flickr30k and use it to encode new ImageNet images [17] and their labels. We then compute the cosine similarity between each image and all sentence labels, selecting the highest-scoring label. As shown in Fig. 6, SUE effectively labels images from the unobserved dataset, despite having only minimal image-text correspondence available during training. Tab. 5 further quantifies these results with a small 50-sample test set, compared to CLIP [60], trained on 400 million pairs. (see App. E.3 for more details).

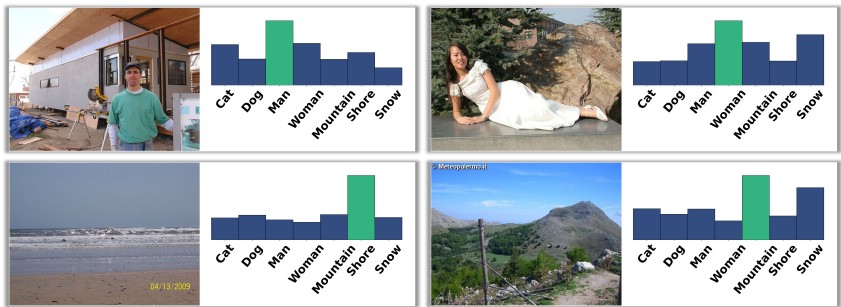

Figure 6: **Zero-shot examples. SUE captures high-level cross-modal semantics.** Classification probabilities by cosine similarity between an ImageNet's images and seven labels, on SUE trained on Flickr30k with 500 pairs. Notably, the images are labeled correctly with only minimal available correspondence during training.

Table 5: **Zero-shot results.** Accuracy results (%) for the zero-shot task. Classification was done using cosine similarity between the ImageNet's images and seven labels, on SUE trained on Flickr30k with 500 pairs. SUE achieves high performance with only very few pairs available during training.

|  | Accuracy |
|---|---|
| CLIP | 99.26 |
| **SUE (ours)** | 88.31 |

## A.4  Emergent Cross-domain Classification

In this experiment, we demonstrate an additional application of universality: emergent cross-domain classification. Specifically, we address the scenario where one domain or modality is labeled, while the other is not, and only very few pairs between them are available. Given two modalities or domains, we train a classifier on the universal embeddings of the labeled domain and apply it to the unlabeled domain, enabling effective knowledge transfer.

Tab. 6 presents classification accuracy results (%) on the Office31 dataset [64], which contains 31 classes and three domains. We used the following two domains: Amazon images and DSLR images. In one experiment, we trained a classifier on the embeddings from the Amazon domain and used it to classify the DSLR domain, and vice versa. SUE achieves high performance with relatively few paired samples, outperforming the MMD baseline (see App. A.1 for more details on this baseline). For additional information on the dataset, its preparation, and classifier, please refer to App. E.4.

Table 6: **Emergent cross-domain classification accuracy results (%).** "Amazon-to-DSLR" indicates that DSLR images were classified using a classifier trained on embeddings from the Amazon domain, and vice versa. SUE significantly outperforms the MMD baseline.

|  | Amazon-to-DSLR | DSLR-to-Amazon |
|---|---|---|
| MMD | 36.78 | 26.33 |
| **SUE (ours)** | **66.02** | **47.40** |

## A.5  mAP Results

In Tab. 7 we provide the mean Average Precision (mAP) results, in the same settings as in Tab. 1. SUE outperforms both contrastive and CSA under this ranking-sensitive metric as well, further validating its effectiveness.

Table 7: **mAP results.** Results with few paired samples on vision-language (🟩🔵), vision-vision (🟩🟩), and tabular-tabular (🟦🟦) datasets from each modality to another: image-to-text (I2T), text-to-image (T2I), edges-to-shoes (E2S), shoes-to-edges (S2E), KL coefficients-to-pixel averages (K2P), pixel-to-KL (P2K); by SUE, Contrastive, and CSA. Using the same small number of pairs, SUE significantly outperforms the popular paired method.

|  | #paired |  | SUE (ours) mAP | Contrastive mAP | CSA mAP |
|---|---|---|---|---|---|
| MSCOCO 🟩🔵 | 100 | I2T | **9.72** | 5.00 | 1.48 |
|  |  | T2I | **9.80** | 4.90 | 1.38 |
| Flickr30k 🟩🔵 | 500 | I2T | **8.91** | 5.80 | 1.67 |
|  |  | T2I | **8.07** | 5.80 | 1.67 |
| Polyvore 🟩🔵 | 500 | I2T | **9.87** | 8.8 | 1.54 |
|  |  | T2I | **9.35** | 9.2 | 1.68 |
| Edges2Shoes 🟩🟩 | 50 | E2S | **7.03** | 5.7 | 1.77 |
|  |  | S2E | **6.99** | 5.1 | 1.88 |
| Handwritten 🟦🟦 | 100 | K2P | **14.38** | 12.60 | 9.42 |
|  |  | P2K | **14.03** | 12.90 | 7.96 |

## A.6  Effect of Different Unimodal Encoders

Tab. 8 analyzes the effect of different pre-trained unimodal encoders on the quality of the shared space. We focus on the MSCOCO dataset, and use text-to-image retrieval results to quantify the quality of the space. In the table, we provide the R@10 result on MSCOCO for all four combinations of four different unimodal encoders. To Dinov2 (ViT-B/14) and SB (MiniLM-L6) which were used in the paper, we add a smaller version of each one - Dinov2 (ViT-s/14) and SB (MiniLM-L3). As expected, higher-capacity unimodal encoders yield better results.

To account for recent advances in text encoders, we additionally evaluate GTR [57], a larger and stronger text encoder, in Tab. 8. Interestingly, its absolute performance is slightly lower. We hypothesize that the Mini-LM was already sufficient given the strength of the vision encoder (DINOv2), or alternatively, that GTR is less aligned with the image representations. In either case, the core effectiveness of our method appears robust to the choice of text encoder.

Table 8: **Encoders comparison.** Text-to-image R@10 results on the MSCOCO dataset using different pre-trained unimodal models. Larger models provide better results.

| Encoders | R@10 |
|---|---|
| Dinov2 (ViT-s/14) + SB (MiniLM-L3) | 30.00 |
| Dinov2 (ViT-B/14) + SB (MiniLM-L3) | 31.50 |
| Dinov2 (ViT-s/14) + SB (MiniLM-L6) | 32.25 |
| Dinov2 (ViT-B/14) + SB (MiniLM-L6)) | **34.25** |
| Dinov2 (ViT-B/14) + GTR (t5-large) | 31.25 |

## A.7  Fig. 1 details

This section outlines the experiment corresponding to the results presented in Fig. 1.

### A.7.1 Similarity of Random Walks on Corresponding Modalities (Fig. 1a)

The objective of this experiment is to evaluate whether the structural relationships among image samples are reflected similarly in their associated text descriptions. To this end, we sampled 1,000 sample batches of image-text pairs from the Flickr30K dataset and constructed the respective random walk matrices for both modalities, $P_I, P_T$. The similarity between random walks was quantified using the normalized Frobenius distance between the corresponding matrices,

$$d(P_I, P_T) = \frac{\|P_I - P_T\|_F}{\|P_I + P_T\|_F}.$$

The blue histogram shows the distribution of distances computed between aligned image-text pairs, while the red histogram reflects distances between randomly shuffled (i.e., unpaired) image and text sets. Although the shuffled matrices preserve statistical properties, they break semantic alignment. The results demonstrate that the distances between corresponding random walks are significantly smaller than those between arbitrary ones, indicating a strong structural similarity. These findings suggest that random walks derived independently from images and texts exhibit a form of universality.

The images and their corresponding captions, used for constructing the demonstration in the figure are depicted in Fig. 7

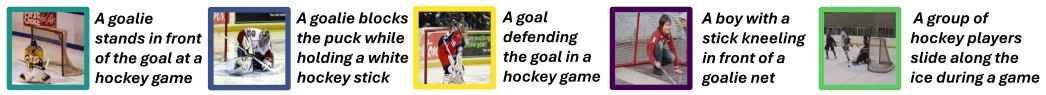

Figure 7: Images and texts used for the construction of the demonstration in Fig. 1a. Colors correspond to the nodes' colors.

### A.7.2 Universal Properties of Spectral Embedding (Fig. 1b)

In this experiment, we investigated whether the eigenfunctions of the Laplace-Beltrami operator exhibit universal properties across modalities. Specifically, we asked whether the eigenfunctions of independently constructed Laplace-Beltrami operators on each unimodal representation, $\psi$ and $\phi$, are similar. To test this, we computed the first three eigenfunctions for each modality in the MSCOCO dataset using SpectralNet [67], and aligned them using CCA on a paired set. Fig. 1b presents the cosine distances between pairs compared to the cosine distances between non-pairs. shows the cosine distances between aligned pairs compared to randomly matched non-pairs. Notably, paired samples are significantly closer in the aligned space, indicating that for a pair $(x, x')$, $\psi(x) \sim \phi(x')$.

### A.8 Extended Comparison to Contrastive with Various #paired

Fig. 8 extends Fig. 5a by including results for the remaining datasets from Tab. 1: Flickr30k, Polyvore, and Edges2Shoes. The figure highlights that SUE effectively leverages unpaired data, enabling a reduction in the number of required paired samples by more than a factor of four on Flickr30k and Edges2Shoes, and by a factor of two on Polyvore.

### A.9 Qualitative Results

**(almost-) text-free text-to-image generation.**  Fig. 9 extends Fig. 4b with additional text-to-image generation results, achieved with almost no text-image correspondence.

**Image retrieval.**  Fig. 10 includes additional qualitative image retrieval examples.

**Text retrieval.**  Fig. 11 includes qualitative texts retrieval examples.

**Edges retrieval.**  Fig. 12 includes qualitative edges retrieval examples on the Edges2Shoes dataset.

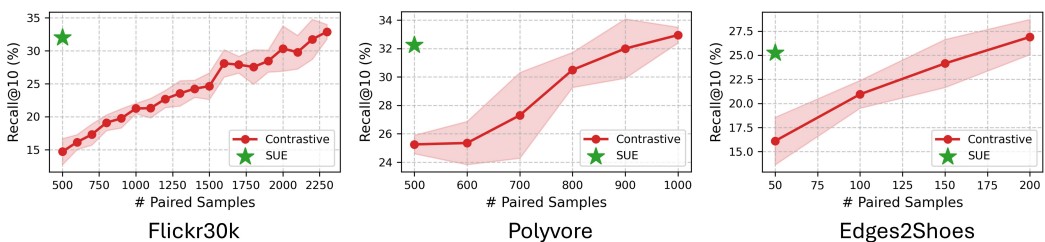

Figure 8: **Contrastive requires much more pairs to achieve similar results as SUE in the weakly-paired regime.** Extension of Fig. 5a. Recall@10 results on Flickr30k, Polyvore, and Edges2Shoes by SUE and Contrastive with various numbers of pairs. SUE exploits unpaired data to outperform contrastive learning when limited pairs are available. Much more pairs are required to achieve similar results with contrastive learning.

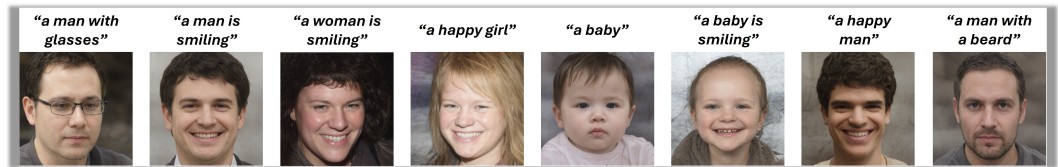

Figure 9: **text-to-image generation with minimal text-image correspondence.** Additional examples.

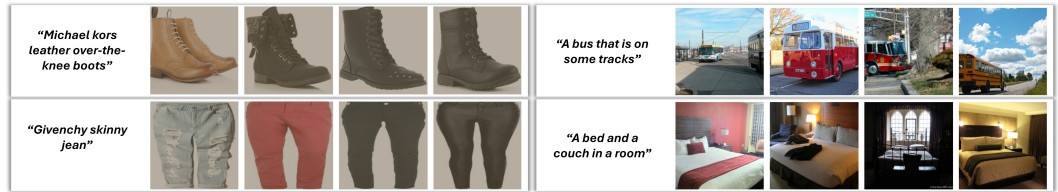

Figure 10: **Additional image retrieval examples.** Right: Polyvore; Left: MSCOCO.

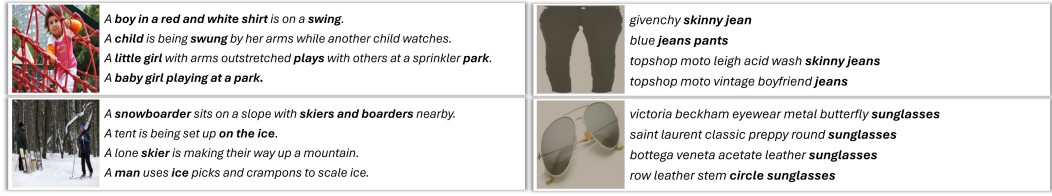

Figure 11: **Text retrieval examples.** Right: Flickr30k; Left: Polyvore.

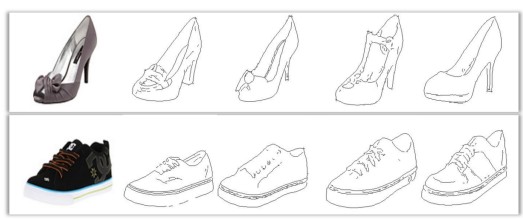

Figure 12: **Edges retrieval examples.** Retrieved edges to shoe queries from the Edges2Shoes dataset.

## A.10 Extended Ablation Study

Tab. 9 extends the CCA and SE + MMD cells in Tab. 2. It includes the retrieval results of these corresponding ablations - CCA (with the same number of paired samples as in SUE), and SE + MMD. The CCA results further support the key role of SE in the universal embedding concept and SUE. In addition, the SE + MMD results show the necessity of CCA in the current version of SUE.

Table 9: **Extended Ablation.** R@10 retrieval results on vision-language (🟩🟦) and vision-vision (🟩🟩) datasets from each modality to another: image-to-text (I2T), text-to-image (T2I), edges-to-shoes (E2S), shoes-to-edges (S2E), note-to-measurement (N2M), measurement-to-note (M2N). SUE results are significantly better, supporting the key role of SE in the universal embedding concept and SUE, as well as the necessity of the CCA step in the current implementation.

|  |  | SUE (ours) | CCA | SE + MMD |
|---|---|---|---|---|
| Flickr30k 🟩🟦 | I2T | 32.00 | 3.75 | 4.75 |
|  | T2I | 32.75 | 4.50 | 5.50 |
| MSCOCO 🟩🟦 | I2T | 34.25 | 7.75 | 3.00 |
|  | T2I | 33.25 | 7.75 | 3.75 |
| Polyvore 🟩🟦 | I2T | 32.25 | 4.00 | 3.75 |
|  | T2I | 32.00 | 4.75 | 4.50 |
| Edges2Shoes 🟩🟩 | E2S | 26.75 | 4.00 | 1.75 |
|  | S2E | 25.50 | 3.25 | 2.50 |

Tab. 10 shows the results of another ablation experiment, where the SE step was replaced with an Auto-Encoder. Notably, the new pipeline considered performs significantly worse compared than SUE, achieving near random results (e.g., 2.75 in R@10, where a random guess would achieve, on mean, 2.5). This again shows the pivotal role of SE in uncovering the universal embedding, as using a different dimensionality reduction method results in an incomparably worse performance.

Table 10: **Spectral Embedding is a key component of universality.** Image-to-text (I2T) and text-to-image (T2I) retrieval results on the Flickr30k dataset with 500 paired samples available during training, using SUE (SE + CCA + MMD), and AE + CCA + MMD.

| #paired | SUE (ours) 500 | | | AE + CCA + MMD 500 | | |
|---|---|---|---|---|---|---|
|  | R@1 | R@5 | R@10 | R@1 | R@5 | R@10 |
| I2T | 5.00 | 20.75 | 32.50 | 0.50 | 1.50 | 2.75 |
| T2I | 5.75 | 23.00 | 33.50 | 0.50 | 1.50 | 2.75 |

## A.11 Ablation Visualization

Fig. 13 visually demonstrates the three steps of SUE (SE, CCA, and MMD) and their combination to achieve a universal embedding. Specifically, the SEs of the two modalities (computed with no paired samples) are globally similar, but not aligned. The CCA step helps to linearly align the two SEs. Then, the MMD step fine-tunes the alignment non-linearly (and with no paired samples). Aligned with Tab. 2, this supports the key role of SE in SUE's pipeline, as the alignment achieved by the CCA and MMD steps is a direct outcome of the SE, and cannot be achieved without it.

# B Modality Gap

Recent research involved the modality gap in multimodal learning [49, 38, 65]. That is, the embeddings of the two modalities of contrastive-based methods are located in two completely separate regions of the embedding space. Some efforts have been made to mitigate this gap [39, 23]. In particular, the modality gap means that the embeddings are not universal, as the modalities lay on different intrinsic manifolds. In contrast, a universal embedding should have no modality gap.

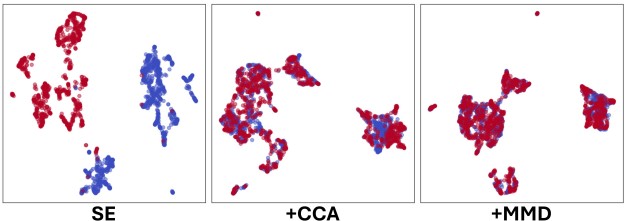

Figure 13: **SUE's UMAP ablation** on Flickr30k. Images in red and texts in blue. The plots represent each of SUE's steps: SE, CCA, and MMD.

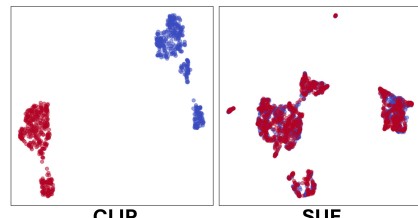

Figure 14: **SUE contains no modality gap.** UMAP embeddings of Flickr30k text and image embeddings by CLIP and SUE. Images in red and texts in blue. Notably, while in the CLIP embedding texts and images lay on two distinct manifolds, SUE places the two modalities on the same manifold.

As visualized in Fig. 14, SUE indeed does not contain a modality gap. The embedding discovered by SUE is a shared manifold of the two modalities (unlike CLIP, for instance). This arises from two core components of our method: CCA produces whitened outputs, ensuring that the representations for each modality are centered at the origin, which implies zero modality gap by definition; and the MMD objective penalizes distributional discrepancies, further reducing any residual mismatch between modalities.

## C Relative CLIP Score

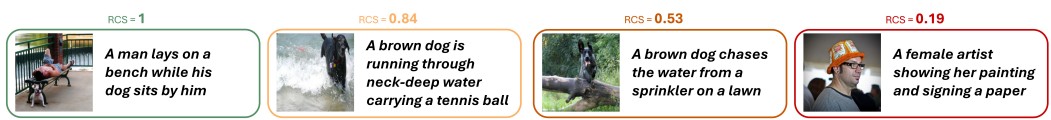

Figure 15: **Relative CLIP Score examples.** The relative CLIP score matches our human intuition of semantic similarity.

The qualitative examples in Fig. 2 and Fig. 4a depict the retrieval weakness in reliably measuring semantic coherence across modalities. We address this by presenting a soft variant of Recall@$k$ for the text-image domain, which we name *Relative CLIP Score* (RCS). CLIP score (CS) is a popular measure of similarity between images [33], but is difficult to interpret. Following this idea, we define the RCS between a pair of an image query and a retrieved text as the CS between them, relative to the CS between the image and its true pair (the RCS between a text query and a retrieved image is defined analogously). This results in a measurement that spans (approximately) between 0 to 1.

Fig. 15 demonstrates RCS in several examples. RCS 1 means that the given image and text are pairs, and as the RCS decreases to 0 the image and pairs become gradually less similar semantically. To enable evaluating the RCS of CLIP, we use a later version of CLIP (ViT-L/14) as the evaluator of RCS.

Tab. 11 shows the RCS of SUE and CLIP on two image-text datasets. Notably, compared to CLIP (trained only on a huge amount of paired samples), SUE achieves about two-thirds performance with no more than 500 paired samples.

Table 11: **Additional Relative CLIP Score results.** RCS results by SUE and CLIP (ViT-L/32). SUE reliably captures global semantic structure while having seen less than 500 pairs during training.

|  | #paired | Flickr30k | |
| --- | --- | --- | --- |
|  |  | I2T | T2I |
| **SUE (ours)** | **500** | 0.667 | 0.638 |
| CLIP | $\sim$ **400M** | 0.971 | 0.973 |

# D    Theoretical Support

In this section, we cite Thm. 21 from [7] to support the claims made in Sec. 4.

**Theorem 1.** *(Bérard et al. [7], Thm. 21) Let $h$ be any metric on $\mathcal{M}$ such that $(1-\epsilon)g \le h \le (1+\epsilon)$, $\epsilon < \epsilon_0$. We assume furthermore that the metrics under consideration have their Ricci curvatures bounded from below by $-(n-1)K^2$ for some constant $K$. There exists constants $\eta_{g,i,K}(\epsilon), 1 \le i \le N_0$, which go to zero with $\epsilon$, such that to any orthogonal basis $\{\psi_j\}$ of eigenfunction of $\Delta_h$ one can associate an orthonormal basis $\{\phi_i\}$ of eigenfunctions of $\Delta_g$ satisfying $\|\phi_i - \psi_i\|_\infty \le \eta_{h,i,K}(\epsilon)$ for $i \le N_0$, where $\|\cdot\|_\infty$ is the sup-norm.*

Intuitively, the result shows that if two embedders $f, g$ of a common manifold $\mathcal{M}$ have bounded distortion and bounded Ricci curvature, then the eigenfunctions of the associated Laplace-Beltrami (or diffusion) operators on $f(\mathcal{M})$ and $g(\mathcal{M})$ are close in the $L_\infty$ sense. For further discussion and explanations we refer to [7].

# E    Technical Details

## E.1    Datasets

We use several datasets of varying modalities to evaluate SUE's performance and the existence of a universal embedding. Tab. 12 details the properties of each dataset.

## E.2    Unimodal Models

As extensively discussed in Sec. 1 and Sec. 4, our first step is extracting meaningful (unimodal) embeddings that capture semantic similarity, using pre-trained unimodal models. The unimodal models used for the different datasets include: DINOv2 (ViT-B/14 distilled) [58], VICReg [2], SBERT (all-MiniLM-L6-v2) [63], the Marqo-FashionSigLIP image encoder[5], and the MPMTS Measurements encoder [41]. Tab. 12 details the models used for each dataset. Tab. 13 details the key properties of each unimodal encoder.

**Feature extraction details.**    For the text modality, we use Sentence-Transformers (e.g., all-MiniLM), which produce fixed-length sentence embeddings via mean pooling as implemented in the library. For the image modality, we extract features using DINOv2, where the [CLS] token serves as the image embedding vector by default.

## E.3    Generation and zero-shot

**Image Generation.**    After obtaining the universal image embeddings from the caption-FFHQ dataset using SUE, we encoded the entire dataset of images with a pre-trained GAN inversion model [77] to obtain the latent representation for all images. Next, we trained a converter model to map the

---

[4]This dataset consists of two handcrafted feature sets derived from images, and therefore can be viewed as a tabular domain.

[5]`https://www.marqo.ai/blog/search-model-for-fashion`

Table 12: **Datasets' properties.** #unpaired indicates the number of unpaired samples used by SUE; #paired indicates the number of paired samples used by SUE. Model 1 refers to the unimodal encoder for the first modality, and Model 2 corresponds to the second modality. More details on the Unimodal encoders can be found in App. E.2

.

| Dataset | Modality 1 | Modality 2 | Model 1 | Model 2 | #unpaired | #paired |
|---|---|---|---|---|---|---|
| Flickr30k | Images | Captions | DINOv2 | SBERT | 27794 | 500 |
| Polyvore | Images | Captions | FashionSigLIP | SBERT | 17190 | 500 |
| MSCOCO | Images | Captions | DINOv2 | SBERT | 8550 | 100 |
| Edges2Shoes | Images | Edges (images) | DINOv2 | VICReg | 12690 | 50 |
| caption-FFHQ | Images | Captions | DINOv2 | SBERT | 16958 | 1000 |
| Office31 | Images | Images | VICReg | DINOv2 | 8000 | 1000 |
| Handwritten[4] | Tabular | Tabular | NA | NA | 1900 | 100 |

Table 13: **Encoders' properties.** For each encoder, we report the backbone architecture, the output dimension, and the number of parameters.

| Model | Backbone | Output Dim. | Params |
|---|---|---|---|
| DINOv2 ViT-B/14 | ViT-B (patch 14) | 768-dim, 257 tokens | ∼86.6M |
| VICReg ResNet-50 | ResNet-50 | 7×7×2048 feature maps | ∼23M |
| FashionSigLIP | SigLIP-based vision-language | 768-dim (vision) | ∼150M |
| SBERT (MiniLM-L6-v2) | MiniLM (6-layer Transformer) | 384-dim sentence | ∼22.7M |

universal image embedding of an input image to its corresponding latent. To achieve this, we used an MSE loss to align the converter's output with the latent obtained from the GAN's encoder.

Once the converter model was trained on the universal image embeddings, we leveraged the universality property as follows: we mapped the text queries to their universal embeddings using SUE, passed them to the converter (which was trained solely on image embeddings), and then fed the output into the GAN's decoder to generate a face image. As shown in Fig. 4b and Fig. 4(c), the converter successfully mapped the universal text embeddings to the same latent space as the images, resulting in meaningful image generation.

For the converter, we used a neural network with two hidden layers, each of size 2048. The training process is configured with a learning rate of $10^{-4}$, a batch size of 512, and an Adam optimizer, running for 600 epochs.

**Zero-shot.** For this experiment, we collected random images of 7 object categories from the ImageNet dataset [17]: cat, dog, man, woman, mountain, shore, snow. Each label was converted into the following sentence: "A photo of a []". For each input image, we computed its universal embedding using SUE, which was trained on Flickr30k and then calculated the cosine similarity between the image embedding and the universal embeddings of all label texts. The label for the image was predicted by selecting the text label that had the highest cosine similarity to the image's universal embedding.

## E.4 Cross-domain Classification

For this experiment, we adjusted the data preparation process. Since the Office31 dataset is relatively small, we augmented the data for each domain using four types of transformations: rotation, flipping, brightness adjustment, and contrast adjustment. This resulted in approximately 30k samples for the Amazon domain and 9k samples for the DSLR domain.

We split the data into a training set (85%) and a test set (15%), ensuring all classes were represented in both splits. For the training set, we created 1k paired samples, consisting of image pairs from the same class, while the remaining training samples were left unpaired, with varying numbers of samples per class across domains.

For unimodal models, we used VICReg [2] encoder for the Amazon domain and DINOv2 [58] for the DSLR domain. After obtaining universal embeddings from SUE on the test set, we constructed a KNN classifier using 5 neighbors on one domain and evaluated it on the other, as explained in App. A.4.

## E.5 Graph Construction

As discussed in Sec. 3, to calculate the Spectral Embedding (SE), a graph affinity matrix is first constructed. Here we detail the graph construction used in SUE.

Given a distance measure $d$ between points, we first compute the $k$-nearest neighbors of each point $x_i$. Let $\{x_{i_1}, x_{i_2}, \ldots, x_{i_k}\}$ represent the $k$-nearest neighbors of $x_i$ under $d$. For each point, we define:

$$\rho_i = \min_j d(x_i, x_{i_j}), \; \sigma_i = \text{median}\{d(x_i, x_{i_j}) \mid 1 \leq j \leq k\}$$

Next, we use a modified RBF kernel to compute the affinity matrix:

$$W_{ij} = \begin{cases} \exp\left(-\frac{(d(x_i, x_j) - \rho_i)^2}{\sigma_i^2}\right) & \text{if } x_j \in \{x_{i_1}, \ldots, x_{i_k}\} \\ 0 & \text{otherwise} \end{cases}$$

Finally, to ensure symmetry, we update $W$ as:

$$W \leftarrow \frac{W + W^\top}{2}.$$

As unimodal models are often trained using cosine similarity (e.g., [63, 58]), we analogously defined $d$ to be the cosine distance:

$$d(x_i, x_j) = 1 - \frac{x_i^T x_j}{\|x_i\| \|x_j\|}$$

## E.6 Training procedure

For the parametric computation of the SE, we adopted the training process presented in [67], and trained a neural network capable of approximating the SE on the training set. Each modality was trained independently, and upon completion, the resulting parametric maps were used to compute the SE for the test points of each modality. For further details on the training process for this parametric map, we refer the reader to [67].

To train the residual network for minimizing the squared MMD loss, we used a publicly available implementation of the MMD loss in PyTorch [6]. The network architecture consists of a single residual connection from the input to the output, without any inner residual connections.

## E.7 Data split and preparation

**Data Split.** For each dataset, we excluded 400 paired samples for evaluation, using the remaining samples for training. To train the parametric SE model, the training set was further divided into a 90% training subset and a 10% validation subset. Similarly, during the training of the MMD network, the training set was partitioned into a 90% training subset and a 10% validation subset.

**Data Preparation.** After projecting the data using the unimodal models, we manipulated the data to include only a few paired samples, resulting in weakly-paired data. Given $n$ training samples, we split them into two portions: a paired portion where the samples remain perfectly aligned across modalities, and an unpaired portion where we introduce controlled noise. For the unpaired portion, we randomly remove 10% of the samples from each modality independently and shuffle their order. This process creates a realistic scenario where only a minimal number of corresponding pairs are available, while the rest are missing or misaligned. Notably, this data preparation procedure is only applied to the training set, as the test set requires properly paired samples to accurately compute cross-modal retrieval metrics and evaluate the model's performance.

---

[6]`https://github.com/yiftachbeer/mmd_loss_pytorch`

## E.8 Hyper-parameters

In this paragraph, we detail all the technical details regarding the different components of our implementation. This includes output dimensions of the different components, network architectures, and hyper-parameters.

**Numeric SE.** For the numeric SE, we constructed the graph as outlined in Sec. E.5, using $k = 100$ for each point. The Laplacian matrix used is the random walk Laplacian, and we selected 10 eigenvectors, which correspond to the output dimension of the SE.

**Parametric SE.** Computing the SE using the parametric approach involves training a neural network. The network architecture consists of an MLP with hidden layers of sizes 4096, 4096, and 1024 for both networks (one per modality). We set the batch size to 4096, and the learning rate to $10^{-4}$ with a decay factor of 0.1, and the training was run for 100 epochs. The optimizer used is Adam, and the learning rate is adjusted using the PyTorch ReduceLROnPlateau scheduler with a patience of 10. We used the same graph construction as used for the numeric SE, outlined in Sec. E.5.

**CCA.** As discussed in Sec. 4.2, the CCA projections are calculated using a small subset of paired samples, with the number of pairs fixed at 600 for all datasets. Fig. 5 presents an experiment demonstrating the impact of the amount of paired data on the results. For these projections, we utilized the CCA implementation from scikit-learn[7], with the number of components set to 8 across all datasets.

**Residual Network (MMD).** To train the residual network, we employed an MLP architecture with hidden layers of size 128, 128, and 128, incorporating a residual connection from the input to the output. The optimizer used is AdamW, with a learning rate set to $10^{-3}$. The network was trained for 100 epochs.

**Auto-Encoder.** To train the Auto-Encoder network, we employed an MLP encoder architecture identical to those of the parametric SE MLP, with a corresponding decoder architecture. The optimizer used is AdamW, with a learning rate set to $10^{-3}$. The network was trained for 100 epochs.

## E.9 OS and Hardware

The training procedures were executed on Rocky Linux 9.3, utilizing Nvidia GPUs including GeForce GTX 1080 Ti and A100 80GB PCIe.

---

[7]`https://scikit-learn.org/stable/modules/generated/sklearn.cross_decomposition.CCA.html`

