# OpenReview forum: "Learning Shared Representations from Unpaired Data"
_NeurIPS.cc/2025/Conference — NeurIPS 2025 poster_

### Official Review · Reviewer_bd36 · 2025-06-29

**Clarity:** 3
**Significance:** 3
**Originality:** 3
**Rating:** 5
**Confidence:** 5

**Summary:**

The authors demonstrate that shared representations can be learned almost exclusively from unpaired data. First, they use pre-trained unimodal encoders to extract feature embeddings from each modality. Then, they use parametric spectral embeddings to reveal universal structures. Finally, they align the modalities linearly via canonical correlation analysis and correct any misalignment using maximum mean discrepancy.

**Questions:**

* The paper would greatly benefit from additional experiments concerning scaling and additional modalities.
* The text encoder seems weak and outdated compared to the vision model, which should be changed.
* The results for other downstream tasks do not seem as convincing and should be better contextualized.
* It is worrying that (Fig. 5c) the method does not scale with additional labels; if the scaling from unpaired data also plateaus, then the proposed approach has reached its limits, while others like CLIP would still scale.

**Ethical Concerns:**

["NO or VERY MINOR ethics concerns only"]

**Final Justification:**

The proposed method is very interesting and has a lot of potential. The additional experiments support the claim that SUE can be applied to other settings. However, as the other reviewers noted, the experimental section is a bit lacking. I suggested to the authors to take a look at the paper "Assessing and Learning Alignment of Unimodal Vision and Language Models" by Zhang et al. (CVPR 2025). Like SUE, SAIL uses two pre-trained encoders from each modality and aligns them. Since both methods use contrastive learning, SAIL may require more paired data than SUE but this needs to be evaluated. Nevertheless, the SAIL paper's experiments are very comprehensive, involving image and text encoders of various sizes and model types which is not given here. The other reviewers also share the same concern as me that the alignment heavily depends on the type of modalities with image and text having strong correlations. Modalities from other domains such as single-cell images and single-cell expressions might not fulfill this requirement. This should be better contextualized in the paper. Overall, I increased my score from 4 to 5 but 4.5 would better reflect my assessment.

**Limitations:**

Limitations stated in the paper.

**Paper Formatting Concerns:**

* Please clarify the assumption k >= r in line 185.
* Why is a residual network only used for Y (196)?
* A universal semantic manifold M is assumed (149). Would the approach work if the modalities are only weakly aligned?
* There is a typo in line 129 ("heat kenrnel").
* Please introduce the full name of CCA when first mentioning it in line 69.
* The claim "proven ability" in 50 needs a reference.

**Quality:**

2

**Strengths And Weaknesses:**

**Strengths**
* The authors address an important problem in multimodal learning, supporting their claims with compelling evidence.
* The theory is clearly explained and well-motivated, which helps readers better understand the proposed method.
* The ablation study complements the main experiments and connects the theoretical and empirical parts of the paper.

**Weakness**
* The experiments were conducted using only image and text modalities. Ideally, more pairs should be included.
* The scaling experiments in Figure 5 should be extended to see if and when the performance plateaus, as in (c).
* The authors use a SOTA vision encoder (DINOv2), but the text encoder (Mini-LM) seems small and outdated.

---

> ### Author Rebuttal · Authors · 2025-07-29
>
> We thank the Reviewer for their thoughtful and encouraging feedback. We are pleased that you found the **problem setting important** and the **theoretical motivation clear and well-explained**. We also appreciate your recognition of the **ablation study** as a meaningful bridge between theory and empirical results. Below, we respond in detail to your comments and suggestions.
>
> **Q.1a: Additional modalities.** We appreciate the Reviewer’s interest in the broader applicability of our method. In response, we have added results on a new multiview benchmark, the Handwritten dataset [1], which can be viewed as a tabular domain. This dataset consists of two handcrafted feature sets derived from images: (1) Karhunen–Loève coefficients (K); and (2) pixel averages over 2x3 windows (P), without the use of foundation models or pretrained embeddings. These experiments are thus fundamentally different from those in the main paper, and they provide evidence that SUE generalizes to non-vision-language settings. The table below extends Tab. 1 from the main paper and shows that SUE significantly outperforms the contrastive baseline on this dataset as well.
>
> | Dataset     | #paired | Direction | SUE R@1 | SUE R@5 | SUE R@10 | Contrastive R@1 | Contrastive R@5 | Contrastive R@10 |
> |-------------|---------|-----------|---------|---------|----------|------------------|------------------|-------------------|
> | Handwritten | 100     | K2P       | 25.50   | 62.00   | 79.00    | 4.80             | 17.00            | 28.00             |
> |             |         | P2K       | 25.00   | 61.75   | 78.00    | 4.80             | 17.80            | 30.50             |
>
> Notably, while our experiments mainly focus on vision-language and vision-vision settings, these domains were selected for their publicly available benchmarks, controllable pairing levels, and intuitive visualization, which support clear comparisons and understanding. We agree that extending SUE to other domains such as medical imaging is a compelling future direction. This requires further and deeper analysis, and we anticipate that SUE's ability to leverage unpaired data will provide clear advantages in those settings.
>
> **Q.1b: Unpaired scaling.** We appreciate the reviewer’s suggestion to extend the scaling experiment in Fig. 5 to observe whether and when performance plateaus. However, Fig. 5b already uses the full available dataset, and unfortunately, no additional unpaired samples are available for this benchmark. Importantly, even at this point, SUE already significantly outperforms the baseline. Notably, performance continues to improve with the amount of unpaired data throughout the observed range. We will clarify this observation in the revised manuscript and view investigating the saturation regime as an important direction for future work with larger or synthetic datasets.
>
> **Q.2: Text encoder.** Thank you for the observation. To address this point, we extended our encoder ablation study in App. A.4 to include a stronger and more recent text encoder, GTR [2]. The results are provided in the table below, extending Tab. 7 from the paper.. Interestingly, absolute performance is slightly lower. We hypothesize that the Mini-LM was already sufficient given the strength of the vision encoder (DINOv2), or alternatively, that GTR is less aligned with the image representations. In either case, the core effectiveness of our method appears robust to the choice of text encoder.
>
> | Encoders                             | R@10  |
> |--------------------------------------|-------|
> | Dinov2 (ViT-B/14) + GTR (t5-large)   | 31.25 |
>
> **Q.3: Downstream task results.** We thank the reviewer for this observation. This paper addresses the question: "What can be achieved when only a small number of pairs are available, alongside large unpaired datasets?" In this setting, standard paired models fall short. While SUE is not intended to compete with models such as CLIP trained on 400M image-text pairs, it offers a compelling alternative for low-pair regimes where such resources are unavailable. In this context, the downstream task performance achieved by SUE is significantly beyond what was previously possible. We will revise the manuscript to better highlight this setting and provide additional context for interpreting downstream results.
>
> **Q.4: Scaling.** We thank the Reviewer for raising this important point. As discussed in previous responses, our work specifically addresses the underexplored and practically relevant setting where only a small number of paired samples are available alongside large amounts of unpaired data. In this scenario, existing contrastive methods like CLIP, which require hundreds of millions of pairs, are not applicable. Instead, our goal is to ask: What is achievable in the absence of large-scale paired datasets? In this regime, SUE offers a strong solution and significantly outperforms existing baselines (see additional baseline in the general comment). We agree that investigating whether performance continues to improve with more unpaired data is an important direction for future work.
>
> **Q.5: $k \geq r$ assumption.** In our method, the original data from each modality lies in $\mathbb{R}^{d_1}$ and $\mathbb{R}^{d_2}$, respectively. These are first reduced to a common dimension $k$ via Spectral Embedding. Then, CCA is applied to project the data from $\mathbb{R}^d$ to a shared latent space of dimension $r$. Since CCA performs a projection onto the top $r$ canonical directions, we require $k \geq r$ to ensure the projection is well-defined. This assumption is standard in CCA-based methods.
>
> **Q.6: MMD network.** The MMD residual network is intended as a final refinement step between two approximately aligned, equi-dimensional distributions. For simplicity and efficiency, we chose to apply the residual network only to $Y$, projecting it to better match $X$. An alternative would be to symmetrically transform both $X$ and $Y$ into a shared space that minimizes the MMD between them. While this could potentially improve alignment, we found the simpler asymmetric approach sufficient in our setting and left such architectural variations for future exploration.
>
> **Q.7: Universal manifold.** The central assumption in our paper is that similarity graphs constructed from unimodal foundation models are sufficiently aligned across different modalities. We support this through extensive experiments in both vision-language and vision-vision domains. As shown in Fig. 2, the random walks and spectral embeddings exhibit strong cross-modal similarity. Furthermore, Sec. 5 empirically validates this assumption by demonstrating that SUE performs effectively across multiple tasks. This assumption is also supported by recent works that report consistent structural alignment between modalities, including text-image pairs [3, 4, 5, 6], as well as more diverse combinations such as audio-text, LiDAR-text, and timeseries-text [7]. While our results affirm the utility of this alignment in current domains, extending SUE to additional modalities is a promising direction for future research and will require further analysis.
>
> **Q.8: proven ability.** Thank you for pointing this out. We will add a citation to [3], which directly demonstrates and discusses the semantic representation capabilities of unimodal models, supporting our claim.
>
> Finally, thank you for pointing out these paper formatting improvements. We will include them in the revised version.
>
> [1] Duin. Multiple Features. UCI Machine Learning Repository, 1998.
>
> [2] Ni et al. Large dual encoders are generalizable retrievers. ACL 2022.
>
> [3] Huh et al. Position: The platonic representation hypothesis. ICML 2024.
>
> [4] Fan et al. Scaling language-free visual representation learning. arXiv 2025.
>
> [5] Moschella, et al. Relative representations enable zero-shot latent space communication. ICLR 2023.
>
> [6] Norelli et al. Asif: Coupled data turns unimodal models to multimodal without training. NeurIPS 2023.
>
> [7] Li et al. Csa: Data-efficient mapping of unimodal features to multimodal features. ICLR 2025.

---

> ### Comment · Reviewer_bd36 · 2025-08-04
>
> Thank you for the detailed rebuttal. I am convinced that the method is interesting and has a lot of potential. The additional experiments support the claim that SUE can be applied to other settings. However, as the other reviewers noted, the experimental section is lacking. I suggest the authors take a look at the paper "Assessing and Learning Alignment of Unimodal Vision and Language Models" by Zhang et al. (CVPR 2025). Like SUE, SAIL uses two pre-trained encoders from each modality and aligns them. Since both methods use contrastive learning, SAIL may require more paired data than SUE but this needs to be evaluated. Nevertheless, the SAIL paper's experiments are very comprehensive, involving image and text encoders of various sizes and model types which is nit given here. As such, I maintain my weak acceptance score.

---

> > ### Author Response · Authors · 2025-08-05
> >
> > We thank the reviewer for finding this work interesting and for their suggestion to compare with SAIL (Zhang et al., CVPR 2025). As the reviewer suggested, we extended Table 1 from the paper, and re-ran SAIL on the three vision-language benchmarks we used in the paper, with the identical number of paired samples used for SUE.
> >
> > For this comparison, we used DINOv2 (ViT-B/14) and SentenceTransformer (MiniLM-L6) as pre-trained encoders for the image and text respectively.
> >
> > As shown below, in the extremely low-pair regime, SUE consistently outperforms SAIL by a large margin, confirming that SUE requires far fewer paired samples to achieve informative retrieval results.
> >
> > | Dataset      | #paired | Direction | SUE R@1 | SUE R@5 | SUE R@10 | SAIL R@1 | SAIL R@5 | SAIL R@10 |
> > |-------------:|--------:|:----------|--------:|--------:|---------:|--------:|--------:|---------:|
> > | MSCOCO       |     100 | I2T       |    5.75 |   21.50 |    34.25 |    0.75 |    4.75 |     7.25 |
> > | MSCOCO       |      | T2I       |    5.25 |   18.25 |    33.25 |    0.75 |    4.50 |     7.00 |
> > | Flickr30k    |     500 | I2T       |    4.25 |   19.75 |    32.00 |    1.00 |    3.75 |     6.50 |
> > | Flickr30k    |      | T2I       |    5.75 |   22.00 |    32.75 |    0.50 |    3.00 |     6.75 |
> > | Polyvore     |     500 | I2T       |    6.00 |   22.75 |    32.25 |   0.50 |    4.25 |     7.25 |
> > | Polyvore     |      | T2I       |    4.75 |   20.75 |    32.00 |    0.75 |    4.00 |     7.50 |
> >
> > We hope this additional comparison satisfies your concerns, and we will add it to the revised version.

---

> ### Comment · Reviewer_bd36 · 2025-08-05
>
> I appreciate the additional last-minute experiments by the reviewer. The improvement over the latest SOTA model SAIL is indeed meaningful and further strengthens the results. In light of this, I change my initial assessment and increase the score to 5 (a more fitting score would be 4.5 as explained below). However, I strongly encourage the authors and appeal to more rigor my including some of the more extensive comparative studies provided in SAIL for a potential camera-ready version (using more model architectures, model sizes and downstream experiments such as few-show classification). I also agree with the other reviewers that the authors should include a limitation section talking about how the model might not work for weakly correlated modality pairs from other domains such as biology (e.g., single-cell images and gene expressions).

---

> > ### Author Response · Authors · 2025-08-06
> >
> > Thank you for your comment and constructive feedback. As you suggested, we will add a set of comparative studies in the camera-ready version. We will also add a Limitations section to note that SUE is best suited to modality pairs with strong semantic overlap, and may require adaptation for weakly correlated domains.

---

### Official Review · Reviewer_M2P3 · 2025-06-30

**Clarity:** 4
**Significance:** 3
**Originality:** 2
**Rating:** 4
**Confidence:** 5

**Summary:**

This paper focuses on a cross-modal representation learning problem with limited paired multimodal data instances and a relatively larger amount of unpaired unimodal data. The proposed method combines CCA and MMD, two existing methods. CCA first alignment between the representations from unimodal encoders, and MMD further refines the alignment.

**Questions:**

1.	Weakness 1: I think a more comprehensive comparison to the referred works is needed, as this paper does not really compare the proposed method to any baselines.
2.	See Weakness 2.
3.	I cannot find the number of parameters of the MMD network. I am skeptical of how the authors trained an MMD network using such a small amount of paired data, especially given that the contrastive baseline does not work.
Overall, this is a strong paper. However, I have concerns about the lack of baseline comparisons and the unresolved issue in Q3. If these are addressed, I will consider updating my score.

**Ethical Concerns:**

["NO or VERY MINOR ethics concerns only"]

**Final Justification:**

Per my comment.

**Limitations:**

No.

Limitations:
The authors mentioned that the limitations are listed in Section 6. However, the section mainly focuses on future works and conclusion, where the discussion on limitations is sparse. I suggest that the authors separate the limitations to another section for clarity.

**Quality:**

3

**Strengths And Weaknesses:**

Strengths:
1.	The paper provides very intuitive explanations to justify why to separate the methodology into 3 stages, especially to justify the random walk stage and the CCA stage.
2.	The paper visualizes the results for the embedding space for a better understanding of the problem and method.
3.	The authors ran an excessive number of experiments on various domains to demonstrate the capability of the proposed method.
Weaknesses:
1.	I believe that the authors missed comparisons and reference to previous works on the same problem: “**when large amounts of unpaired data are available alongside only a very small number of paired samples**”, which contradicts the authors’ claim: “**To the best of our knowledge, this work is the first to learn shared representations from almost exclusively unpaired data.**”
The authors should compare their method with at least the most recent work cited below. [3] also leverages CCA to find a multimodal representation space with limited paired data, which I think is the same setting and similar methodology as this paper.

Reference:
[1]  Moschella, Luca, et al. "Relative representations enable zero-shot latent space communication." International Conference on Learning Representations (2023)
[2] Norelli, Antonio, et al. "Asif: Coupled data turns unimodal models to multimodal without training." Advances in Neural Information Processing Systems 36 (2023).
[3] Li, Po-han, et al. "CSA: Data-efficient Mapping of Unimodal Features to Multimodal Features". International Conference on Learning Representations (2025)

2.	There is no ablation study on how changing the unimodal encoders affects the performance on the various tasks and the intuition of when to use which encoder.
Minor Weaknesses:
1.	The caption of Fig. 1 does not indicate that the retrieval result is by SUE.
2.	“he Marqo-FashionSigLIP image encoder” in Section D.
3.	The figures are not sorted in the given numbers in the appendix (Fig. 14 comes earlier than Fig. 12 and Fig. 13).

---

> ### Author Rebuttal · Authors · 2025-07-29
>
> We thank the Reviewer for their detailed and constructive review. We are glad that you appreciated the **intuitive justifications** provided for our three-stage methodology. We also value your positive remarks on our **visualizations**, which aim to offer clearer insight into the embedding space, and on our **extensive experimental evaluation** across diverse domains, which supports the effectiveness of our approach. Below, we address your comments, add new results, and clarify specific points.
>
> **Q.1.** We thank the reviewer for pointing out these relevant and timely works. In response, we have incorporated a comparison to CSA [3], which builds upon and improves over ASIF [2], itself based on [1]. Accordingly, we include all three works in the related work section and compare directly against CSA, the strongest among them.
>
> As shown in our updated results below, SUE significantly outperforms CSA in the extremely low-pair regime we focus on. This highlights a key distinction: while CSA leverages limited paired data, it fails to operate in the setting we address, where the number of available pairs is extremely small, and unpaired data plays a central role.
>
> Therefore, although these prior works explore related goals, to the best of our knowledge, ours is the first to demonstrate successful shared representation learning from almost exclusively unpaired data. We will emphasize this distinction and the expanded comparison in the revised version.
>
> The table below extends Tab. 1 in the main paper and compares our method (SUE), contrastive training, and CSA under the same limited pairing regime. Notably, while CSA effectively reduces the number of pairs needed to match CLIP, it is unable to operate in the extreme low-pair regime addressed by our paper. Specifically, when the number of pairs $n$ falls below the input dimension $d = \min\{d_1, d_2\}$, CSA collapses. To enable a fair comparison, we use the minimal feasible number of pairs for CSA ($n=d$), which is still larger than the number used by our method (SUE). Even in this more favorable setting for CSA, our method (SUE) significantly outperforms it, highlighting the strength of our approach in this regime, and in leveraging unpaired data.
>
> We will include these results in the revised version of the manuscript.
>
> | Dataset      | #paired | Direction | SUE R@1 | SUE R@5 | SUE R@10 | Contrastive R@1 | Contrastive R@5 | Contrastive R@10 | CSA R@1 | CSA R@5 | CSA R@10 |
> |--------------|---------|-----------|---------|---------|----------|------------------|------------------|-------------------|---------|---------|----------|
> | MSCOCO       | 100     | I2T       | 5.75    | 21.50   | 34.25    | 1.50             | 8.50             | 13.00             | 0.00    | 1.25    | 3.00      |
> |              |         | T2I       | 5.25    | 18.25   | 33.25    | 0.80             | 5.80             | 12.20             | 0.00    | 1.00    | 2.25      |
> | Flickr30k    | 500     | I2T       | 4.25    | 19.75   | 32.00    | 3.00             | 9.50             | 16.20             | 0.25    | 1.25    | 2.50      |
> |              |         | T2I       | 5.75    | 22.00   | 32.75    | 2.50             | 9.80             | 15.00             | 0.25    | 0.75    | 2.75      |
> | Polyvore     | 500     | I2T       | 6.00    | 22.75   | 32.25    | 3.20             | 13.80            | 22.50             | 0.25    | 1.25    | 2.25      |
> |              |         | T2I       | 4.75    | 20.75   | 32.00    | 4.00             | 11.50            | 23.00             | 0.25    | 1.00    | 3.25      |
> | Edges2Shoes  | 50      | E2S       | 4.00    | 16.00   | 25.25    | 1.00             | 5.50             | 14.00             | 0.25    | 1.50    | 2.75      |
> |              |         | S2E       | 3.50    | 17.00   | 27.00    | 0.80             | 6.00             | 12.80             | 0.25    | 1.50    | 3.00      |
>
> **Q.2.** The paper includes an ablation study on how changing the unimodal encoders affects the retrieval performance, in App. A.4. It is referred to from Sec. 5.1.
>
> **Q.3.** The architecture details for all components, including the MMD network, are provided in Appendix D.8. As discussed in Section 4.1, the MMD network is trained entirely on unpaired data, which is available in large quantities. It does not rely on paired supervision, so the limited number of pairs is not a limiting factor here. However, MMD alignment (especially using a residual network) assumes that the input distributions are already roughly aligned. For this reason, the MMD step acts only as a final fine-tuning stage, building on the alignment achieved by the previous pipeline components. As shown in Table 2 and Appendix A.1, MMD alone fails without this prior structure, reinforcing its role as a lightweight refinement rather than a standalone solution.
>
> Finally, thank you for pointing out these minor issues. We will address them in the revised version.
>
> [1] Moschella, et al. Relative representations enable zero-shot latent space communication. ICLR 2023.
>
> [2] Norelli et al. Asif: Coupled data turns unimodal models to multimodal without training. NeurIPS 2023.
>
> [3] Li et al. Csa: Data-efficient mapping of unimodal features to multimodal features. ICLR 2025.

---

> > ### Comment · Reviewer_M2P3 · 2025-08-07
> > **Addressed the comments.**
> >
> > I appreciate the detailed reply. I think the authors did a detailed comparison of CSA and SUE under very limited paired data for retrieval tasks. I will raise my score accordingly. A very nitpicky comment: There’s the possibility to stabilize CSA under limited pairs of data lower than the dimension with regularization, but of course that is beyond the scope of the paper, and I think SUE focuses on a very interesting and special use case. I will increase my score.

---

> > > ### Author Response · Authors · 2025-08-08
> > >
> > > Thank you very much for your support and help in improving this paper.

---

### Official Review · Reviewer_MCZ3 · 2025-07-01

**Clarity:** 2
**Significance:** 3
**Originality:** 2
**Rating:** 4
**Confidence:** 5

**Summary:**

The paper aims to learn multi-modal representations from unpaired data. Current methods are trained on large amounts of paired data, which is often difficult to obtain. This work learns shared embeddings largely from unpaired data using spectral embeddings of random walk matrices built separately for each modality. The method is validated on vision-language and vision-vision datasets.

**Questions:**

1. The method is validated primarily on vision-language tasks with datasets that are carefully annotated paired data. It is unclear whether the approach would generalize to other modality pairs with weaker or noisier alignment. Further, the method is used for vision-language datasets which already have large amounts of paired data. What would be interesting is if this could be validated in domains where such aligned pairs are hard to obtain such as medical imaging, etc.
2. The final MMD-based alignment step assumes that the distributions are already well-aligned after CCA. If this initial alignment is poor, MMD alone may not be sufficient to close the gap between modalities without paired supervision. How would this be mitigated?
3. More details about how these unpaired datasets were constructed are needed.
4. For retrieval, mAP should also be reported which considers the ranking of the retrieved examples.
5. The paper claims ‘An order of magnitude more pairs are required to achieve similar results with contrastive learning’. Was any example mining for contrastive loss done? Would the result still hold if informative sampling is done?

**Ethical Concerns:**

["NO or VERY MINOR ethics concerns only"]

**Final Justification:**

The rebuttal addressed most of my concerns and included additional evaluation. So I have raised my scores. However, I am still not convinced about W1, W2. While this may work for certain modalities that are semantically aligned, many modalities such as audio-video may capture complementary information, and this method may not work.

**Limitations:**

Several important limitations as highlighted in detail in Weaknesses and Questions sections.
1. Mainly related to crucial assumptions that the method is based on
2. Need more extensive evaluation, comparison with the state-of-the-art and ablation studies

**Quality:**

2

**Strengths And Weaknesses:**

Strengths:
1. The paper is well-written and addresses an important topic. The motivation behind the problem is explained clearly.
2. Provided ablation studies and visualization for validation on vision-language and vision-vision datasets.

Weaknesses:
1. The approach assumes that semantic similarity graphs (for example as shown in Fig 2), derived from unimodal foundation models are sufficiently aligned across different modalities. In practice, text and images can emphasize different semantic aspects. Also, the text caption is subjective to the annotator. This crucial assumption may not always hold.
2. On a similar note, in lines 148-150, the approach assumes the existence of a single unified latent semantic manifold shared across modalities. Realistically, the semantic structure captured by different modalities (e.g., images vs. text) may differ significantly, violating this assumption.
3. The argument that similar random walks imply aligned eigenfunctions is presented without rigorous guarantees. Stronger theoretical analysis is required to validate this critical assumption.
4. The empirical evidence for the approach is drawn largely from vision-language data and some vision-vision data with curated and strongly paired datasets. It is unclear how well the method would generalize to other modality pairs with weaker or noisier semantic alignment. Further, the method is used for vision-language datasets which already have large amounts of paired data. What would be interesting is if this could be validated in domains where such aligned pairs are hard to obtain such as medical imaging, etc.
5. My major concern is the lack of extensive evaluation. The method is only compared against Contrastive loss and no other state-of-the art methods or losses.
6. An important ablation is needed. How much % of unpaired data is needed to roughly get similar performance as compared to fully paired data. Performance plotted across % of unpaired data would give us better insights about how well this method works.

---

> ### Author Rebuttal · Authors · 2025-07-29
>
> We thank the Reviewer for their insightful review. We are pleased that you found the paper to be **well-written** and addressing an **important topic**. We appreciate your recognition of the **clear motivation** behind our problem setting, as well as the value of our **ablation studies and visualizations**, which aim to validate our approach across both vision-language and vision-vision tasks. Below, we respond to your comments and provide further results and clarifications where needed.
>
> **W.1.** The assumption made in the paper, that similarity graphs, derived from unimodal foundation models are sufficiently aligned across different modalities, is supported in the paper with extensive experiments in the vision-language and vision-vision domains. As shown in Fig. 2, the random walks and spectral embeddings exhibit strong cross-modal similarity. Furthermore, Sec. 5 empirically validates this assumption by demonstrating that SUE performs effectively across multiple tasks. This assumption is also supported by recent works that report consistent structural alignment between modalities, including text-image pairs [1, 2, 3, 4], as well as more diverse combinations such as audio-text, LiDAR-text, and timeseries-text [5]. Notably, while text and images may each capture modality-specific nuances, prior work demonstrates that their global semantic structures remain well-aligned.
>
> **W.2.** Similarly to our answer above, perhaps surprisingly, we show that such a unified latent semantic manifold exists to some extent. While text and images may each capture modality-specific nuances, they still describe the same semantic objects, and thus share their global semantic structure. This assumption is supported by previous work [1, 2, 3, 4, 5].
>
> **W.3.** Thank you for this comment. In Section 4.1, we support the argument that similar random walks imply aligned eigenfunctions by referencing Theorem 21 from [6]. Specifically, the result shows that if two embeddings $f,g$ of a common manifold $\mathcal{M}$ have bounded distortion and bounded Ricci curvature, then the eigenfunctions of the associated Laplace-Beltrami (or diffusion) operators on $f(\mathcal{M})$ and  $g(\mathcal{M})$ are close in the $L_{\infty}$ sense. For completeness, we will add the theorem to the appendix.
>
> This provides a rigorous justification for our assumption in the setting where the observed data lie on geometrically similar manifolds. While these assumptions may not hold perfectly in all practical cases, they offer a strong theoretical foundation for why alignment via spectral structure is both plausible and effective, especially when supported by empirical evidence such as the graph similarities shown in Fig. 2.
>
> **W.4.** We appreciate the Reviewer’s interest in the broader applicability of our method. In response, we have added results on a new multiview benchmark, the Handwritten dataset [7], which can be viewed as a tabular domain. This dataset consists of two handcrafted feature sets derived from images: (1) Karhunen–Loève coefficients (K); and (2) pixel averages over 2x3 windows (P), without the use of foundation models or pretrained embeddings. These experiments are thus fundamentally different from those in the main paper, and they provide evidence that SUE generalizes to non-vision-language settings. The results (see table in W.5 below) show that SUE significantly outperforms the contrastive baseline on this dataset as well.
>
> Notably, while our experiments mainly focus on vision-language and vision-vision settings, these domains were selected for their publicly available benchmarks, controllable pairing levels, and intuitive visualization, which support clear comparisons and understanding. We agree that extending SUE to other domains such as medical imaging is a compelling future direction. This requires further and deeper analysis, and we anticipate that SUE's ability to leverage unpaired data will provide clear advantages in those settings.
>
> **W.5.** Following the Reviewer’s suggestion, we have compared SUE to an additional baseline below. Specifically, we have added a comparison to CSA (proposed by Reviewer M2P3). CSA [5] is a recently introduced method designed to match CLIP performance while reducing reliance on paired data. The table below extends Tab. 1 in the main paper and compares our method (SUE), contrastive training, and CSA under the same limited pairing regime. Importantly, SUE significantly outperforms this new baseline in the extremely low-pair regime we focus on. We hope that the additional comparison provides better context for SUE's performance.
>
> Notably, while CSA effectively reduces the number of pairs needed to match CLIP, it is unable to operate in the extreme low-pair regime addressed by our paper. Specifically, when the number of pairs $n$ falls below the input dimension $d = \min\{d_1, d_2\}$, CSA collapses. To enable a fair comparison, we use the minimal feasible number of pairs for CSA ($n=d$), which is still larger than the number used by our method (SUE). Even in this more favorable setting for CSA, our method (SUE) significantly outperforms it, highlighting the strength of our approach in this regime, and in leveraging unpaired data.
>
> | Dataset     | #   | | SUE R@1 |SUE R@5  |SUE R@10 | Contrastive R@1 |Contrastive  R@5  |Contrastive  R@10 | CSA R@1 | CSA R@5  | CSA R@10 |
> |----------|-----|-----|------|-----|-----|------|-----|-----|--------|-----|-----|
> | MSCOCO   |100  | I2T | 5.75 |21.5 |34.3| 1.50 | 8.5 |13.0 |   0.00 |1.25 |3.00 |
> |          |     | T2I | 5.25 |18.3 |33.3| 0.80 | 5.8 |12.2 |   0.00 |1.00 |2.25 |
> | Flickr30k|500  | I2T | 4.25 |19.8 |32.0| 3.00 | 9.5 |16.2 |   0.25 |1.25 |2.50 |
> |          |     | T2I | 5.75 |22.0 |32.8| 2.50 | 9.8 |15.0 |   0.25 |0.75 |2.75 |
> | Polyvore |500  | I2T | 6.00 |22.8 |32.3| 3.20 |13.8 |22.5 |   0.25 |1.25 |2.25 |
> |          |     | T2I | 4.75 |20.8 |32.0| 4.00 |11.5 |23.0 |   0.25 |1.00 |3.25 |
> | E2Shoes  | 50  | E2S | 4.00 |16.0 |25.3| 1.00 | 5.5 |14.0 |   0.25 |1.50 |2.75 |
> |          |     | S2E | 3.50 |17.0 |27.0| 0.80 | 6.0 |12.8 |   0.25 |1.50 |3.00 |
> | Handwritten|100  | K2P |25.50 |62.0 |79.0| 4.80 |17.0 |28.0 |   4.25 |12.2 |17.5 |
> |          |     | P2K |25.00 |61.8 |78.0| 4.80 |17.8 |30.5 |   3.50 |9.00 |15.0 |
>
>
> **W.6.** We thank the reviewer for raising this important point. The tradeoff between the amount of unpaired and paired data may be extracted from in Fig. 5b and Tab. 1. For instance, on the Flickr30k image retrieval task, the contrastive baseline achieves an R@10 of approximately 15 using 500 paired examples. SUE reaches comparable performance using around 15,000 unpaired samples. Importantly, as the amount of unpaired data increases, SUE's performance continues to improve, without requiring additional paired supervision. We will clarify this insight in the revised manuscript.
>
> **Q.1.** Please see our response to **W.4**.
>
> **Q.2.** We agree with the reviewer that the MMD-based alignment step is most effective when the modality-specific embeddings are already roughly aligned. As shown in our experiments (e.g., Tab. 1 and Tab. 2), this initial alignment is relatively strong, which allows the final MMD step to refine rather than create alignment. We also confirm the reviewer’s intuition that MMD alone is insufficient to bridge the modalities without prior alignment: our ablation study in Tab. 2 demonstrates that using MMD without the spectral embedding and CCA steps results in significantly worse performance.
>
> **Q.3.** Please see "Data Preparation" paragraph in App. D.7.
>
> **Q.4.** We thank the reviewer for the suggestion. We have computed mean Average Precision (mAP) results, and summarized them in the table below. SUE outperforms the baselines under this ranking-sensitive metric as well, further validating the effectiveness of our approach.
>
> | Dataset       | # | | **SUE** mAP  | Contrastive mAP| CSA mAP|
> |------------|----|-----|------|------|------|
> | MSCOCO     |100 | I2T | **9.72** | 5.00 | 1.48 |
> |            |    | T2I | **9.80** | 4.90 | 1.38 |
> | Flickr30k  |500 | I2T | **8.91** | 5.80 | 1.67 |
> |            |    | T2I |**8.07**| 5.80 | 1.67 |
> | Polyvore   |500 | I2T |**9.87** | 8.80 | 1.54 |
> |            |    | T2I |**9.35**| 9.20 | 1.68 |
> | E2Shoes    | 50 | E2S |**7.03** | 5.70 | 1.77 |
> |            |    | S2E |**6.99** | 5.10 | 1.88 |
> | Handwritten  |100 | K2P |**14.38** |12.60 | 9.42 |
> |            |    | P2K |**14.03**|12.90 | 7.96 |
>
>
> **Q.5.** We thank the reviewer for raising this point. If we understand correctly, the question refers to the use of informative example mining to improve the performance of contrastive learning. While this is a powerful technique in fully supervised settings, it is not applicable in our scenario. Our work explicitly focuses on the low-pair regime, where only a small, fixed set of aligned pairs is available, and additional pair selection or mining is not possible.
>
> This setting reflects real-world constraints in weakly supervised multimodal learning, where collecting or identifying informative pairs is either expensive or infeasible. Therefore, the contrastive baselines we compare against operate under the same limited supervision, and our claim that contrastive learning requires an order of magnitude more pairs holds under this constraint.
>
> [1] Huh et al. Position: The platonic representation hypothesis. ICML 2024.
>
> [2] Fan et al. Scaling language-free visual representation learning. arXiv 2025.
>
> [3] Moschella, et al. Relative representations enable zero-shot latent space communication. ICLR 2023.
>
> [4] Norelli et al. Asif: Coupled data turns unimodal models to multimodal without training. NeurIPS 2023.
>
> [5] Li et al. Csa: Data-efficient mapping of unimodal features to multimodal features. ICLR 2025.
>
> [6] Bérard et al. Embedding riemannian manifolds by their heat kernel. GAFA 1994.
>
> [7] Duin. Multiple Features. UCI Machine Learning Repository, 1998.

---

> > ### Comment · Reviewer_MCZ3 · 2025-08-05
> > **Official Comment**
> >
> > The rebuttal addressed most of my concerns and included additional evaluation. So I have raised my scores. However, I am still not completely convinced about W1, W2. While this may work for certain modalities that are semantically aligned, many modalities such as audio-video may capture complementary information, and this method may not work. I would suggest mentioning such limitations in the paper/supplemental.

---

> > > ### Author Response · Authors · 2025-08-06
> > >
> > > Thank you for your comment. We agree that it is most natural to apply SUE when the two modalities share strong semantic overlap (e.g., image–text). We will add a Limitations section to the revised manuscript discussing the challenges in modalities that are less semantically aligned.

---

### Official Review · Reviewer_bSLV · 2025-07-02

**Clarity:** 4
**Significance:** 4
**Originality:** 4
**Rating:** 5
**Confidence:** 3

**Summary:**

This paper introduces a method for learning a shared embedding for multiple modalities, such as text and image, using primarily unpaired data. The core idea is to first learn unimodal representations for each modality separately using spectral embeddings, which are argued to effectively preserve the global manifold structure. Then these independently learned manifolds are aligned into a shared space using MMD and a small set of paired samples via CCA. The primary contribution is a framework that significantly reduces dependency on large-scale paired data.

**Questions:**

1.  **Impact of Pre-trained Encoders:** Could you please clarify exactly how features were extracted from the pre-trained backbones (e.g., `[CLS]` token vs. averaged features)? This introduces a hidden confounder that may "pre-homogenize" the features between modalities. E.g. filter out features from the backgrounds in images (which also tend not appear in the captions). Please discuss the potential confounding effect of using representations that were part of a pre-training objective.
I would expect representations of images/videos to have a higher information density than text (especially captions), in which case the global manifold structure across modalities may not be as alignable - does this undercut the proposed method?
2.  **Baseline Comparisons:** Could you provide comparisons against established semi-supervised or weakly-supervised multimodal learning methods? This is critical for assessing the paper's contribution, and my score would increase significantly if SUE proves competitive.
3.  **Robustness of "No Modality Gap" Claim:** The claim that SUE has no modality gap (Appendix B) is quite interesting. Could it be highlighted more in the main paper? Is this observation robust across different datasets and embedding dimensions, or is it specific to the UMAP visualization shown in Figure 14? Generally, there should be more (of the extensive) results in the main paper, rather than relegated to the appendix.

**Ethical Concerns:**

["NO or VERY MINOR ethics concerns only"]

**Final Justification:**

I have raised my score to reflect the authors' strong rebuttal, which directly addressed several of my concerns by adding a several new empirical results and discussion. However, I believe fully integrating the additional discussion, clarifications, and results may benefit from a resubmission. Nevertheless, the work is very promising.

**Limitations:**

No. The limitations should be discussed more thoroughly - e.g. inherent potential trade-offs between preserving global structure and fidelity.

**Paper Formatting Concerns:**

None.

**Quality:**

3

**Strengths And Weaknesses:**

### Strengths

**Originality and Significance:** The paper tackles the highly significant problem of reducing the need for large-scale paired data in multimodal representation learning.

**Methodological Soundness:** The design of the method is well-motivated, and explained clearly. The ablation studies presented in the appendix are a particular strength, clearly demonstrating how each component helps. Extensive experiments across multiple datasets and tasks provide strong empirical support for the method's effectiveness.

### Weaknesses

**Insufficient Baseline Comparisons:** The most critical weakness is the lack of adequate baseline comparisons. The comparison only comparison (in the main paper), is a naive contrastive baseline, making it impossible to judge the practical utility and trade-offs of the method. While the authors argue no direct baselines exist for this setting, several semi- or weakly-supervised methods (mentioned in the related work), would help contextualize the proposed method. Some discussion of existing methods for aligning representations such as https://openreview.net/forum?id=eimAJqoIWt&noteId=eimAJqoIWt or https://openreview.net/forum?id=l2izo0z7gu
On a related note, the choice of the autoencoder as a non-spectral baseline in the ablation study is not well-justified - why not a contrastive method (e.g. SimCLR)?

**Coarse-Grained Alignment:** The qualitative results, particularly in the concept arithmetic experiments (Figure 4c), suggest that the alignment between modalities is rather coarse, failing to preserve fine-grained details like facial identity. This may be an inherent trade-off of the chosen approach; by prioritizing global manifold structure via spectral embeddings, the method may naturally lose local, fine-grained information. This trade-off should be investigated more thoroughly.

### Minor Issues

- l. 69: the abbreviation CCA is never defined, unlike MMD and SE.
- l. 129: "kenrnel" -> "kernel"

---

> ### Author Rebuttal · Authors · 2025-07-29
>
> We thank the Reviewer for their thoughtful and positive feedback. In particular, we are encouraged by the acknowledgment of the **importance and originality** of the problem we tackle, and the **clarity and motivation** underlying our methodology. We also appreciate your recognition of the **extensive empirical validation** of our approach, as well as the **insightful ablation studies** that demonstrate the contribution of each component. Below, we address your comments in detail, add new results, and clarify the points you raised.
>
> **W.1**
>
> **Baseline comparison.** Following the Reviewer’s suggestion, we have compared SUE to an additional baseline below. Specifically, we have added a comparison to CSA (proposed by Reviewer M2P3). CSA [1] is a recently introduced method designed to match CLIP performance while reducing reliance on paired data. The table below extends Tab. 1 in the main paper and compares our method (SUE), contrastive training, and CSA under the same limited pairing regime. Importantly, SUE significantly outperforms this new baseline in the extremely low-pair regime we focus on. We hope that the additional comparison provides better context for SUE's performance.
>
> Notably, while CSA effectively reduces the number of pairs needed to match CLIP, it is unable to operate in the extreme low-pair regime addressed by our paper. Specifically, when the number of pairs $n$ falls below the input dimension $d = \min\{d_1, d_2\}$, CSA collapses. To enable a fair comparison, we use the minimal feasible number of pairs for CSA ($n=d$), which is still larger than the number used by our method (SUE). Even in this more favorable setting for CSA, our method (SUE) significantly outperforms it, highlighting the strength of our approach in this regime, and in leveraging unpaired data.
>
> We will include these results in the revised version of the manuscript.
>
> | Dataset      | #paired | Direction | SUE R@1 | SUE R@5 | SUE R@10 | Contrastive R@1 | Contrastive R@5 | Contrastive R@10 | CSA R@1 | CSA R@5 | CSA R@10 |
> |--------------|---------|-----------|---------|---------|----------|------------------|------------------|-------------------|---------|---------|----------|
> | MSCOCO       | 100     | I2T       | 5.75    | 21.50   | 34.25    | 1.50             | 8.50             | 13.00             | 0.00    | 1.25    | 3.00      |
> |              |         | T2I       | 5.25    | 18.25   | 33.25    | 0.80             | 5.80             | 12.20             | 0.00    | 1.00    | 2.25      |
> | Flickr30k    | 500     | I2T       | 4.25    | 19.75   | 32.00    | 3.00             | 9.50             | 16.20             | 0.25    | 1.25    | 2.50      |
> |              |         | T2I       | 5.75    | 22.00   | 32.75    | 2.50             | 9.80             | 15.00             | 0.25    | 0.75    | 2.75      |
> | Polyvore     | 500     | I2T       | 6.00    | 22.75   | 32.25    | 3.20             | 13.80            | 22.50             | 0.25    | 1.25    | 2.25      |
> |              |         | T2I       | 4.75    | 20.75   | 32.00    | 4.00             | 11.50            | 23.00             | 0.25    | 1.00    | 3.25      |
> | Edges2Shoes  | 50      | E2S       | 4.00    | 16.00   | 25.25    | 1.00             | 5.50             | 14.00             | 0.25    | 1.50    | 2.75      |
> |              |         | S2E       | 3.50    | 17.00   | 27.00    | 0.80             | 6.00             | 12.80             | 0.25    | 1.50    | 3.00      |
>
> **References.** We thank the reviewer for pointing us to relevant references. Both will be cited and discussed in the revised related work section. However, similar to the other methods already mentioned in the related work, these approaches assume access to additional supervision or large-scale paired training data. For example, [2] incorporates class-level labels into the alignment process, whereas our setting is label-free. [3] depends on multimodal foundation models trained on hundreds of millions of paired examples, which falls outside the minimal-pair regime we consider. As such, while informative, these methods are not directly comparable to SUE.
>
> **AE ablation.** Our framework operates on pre-computed semantic embeddings (e.g., from Sentence-Transformers or DINOv2), not raw input data. In this setting, applying SimCLR or related contrastive methods is non-trivial: these methods are designed for raw data and rely heavily on stochastic augmentations to define positive/negative pairs, an approach that does not translate naturally to pre-extracted embedding spaces. Moreover, the contrastive paradigm assumes access to meaningful augmentations or label structure to guide the loss. In contrast, the autoencoder serves as a natural and neutral baseline in the embedding space, offering a non-spectral method that reconstructs input embeddings without introducing alignment or graph structure. This allows us to isolate and highlight the specific contribution of our spectral learning component.
>
> **W.2. Coarse-Grained Alignment:** We thank the reviewer for this insightful observation. Our work addresses the question: "What can be achieved when only a small number of cross-modal pairs are available alongside large unpaired datasets?" In this low-pair regime, standard paired models are inapplicable or ineffective. SUE provides a principled alternative that leverages the intrinsic structure of each modality for alignment, achieving downstream performance way beyond what was previously possible.
>
> We agree with the Reviewer that our approach, by prioritizing global manifold structure through spectral embeddings, may naturally trade off some fine-grained detail, in favor of broader semantic alignment. We view this as an inherent and interesting property of our method, and believe that a deeper investigation of this trade-off between global alignment and local fidelity is a valuable direction for future work. We will revise the manuscript to clarify this point and provide better context for interpreting these qualitative results.
>
> **Q.1**
>
> **Feature extraction details.** For the text modality, we use Sentence-Transformers (e.g., all-MiniLM), which produce fixed-length sentence embeddings via mean pooling as implemented in the library. For the image modality, we extract features using DINOv2, where the [CLS] token serves as the image embedding vector by default. We will add these to the technical details appendix.
>
> **On the potential confounding effect.** We acknowledge that pre-trained encoders introduce semantic biases that may facilitate cross-modal alignment, as reflected in Fig. 2. However, rather than viewing this as a hidden confounder, we consider this semantic preprocessing a deliberate and essential design choice. By embedding modalities into meaningful feature spaces, SUE can effectively leverage unpaired data for alignment. Attempting alignment from raw pixels and tokens without such semantic abstraction would demand far more paired supervision, contradicting our goal of minimal pairing.
>
> **Information density and alignability.** Regarding the comment that images contain higher information density than captions, our method capitalizes on the semantic abstraction of these encoders. Both DINOv2 and Sentence-Transformers distill inputs into embeddings, emphasizing relevant semantic content while reducing irrelevant details such as background noise. This abstraction actually enhances alignability, as supported by the similar graph structures observed in Fig. 2, demonstrating that both modalities capture analogous semantic relationships conducive to spectral alignment.
>
> **Q.2.** Please see our response to **W.1**.
>
> **Q.3 Modality gap.** Thank you for your interest in our discussion of the modality gap. This finding is not limited to the UMAP visualization in Fig. 14, it holds consistently across datasets and embedding dimensions. It arises from two core components of our method: (1) CCA produces whitened outputs, ensuring that the representations for each modality are centered at the origin, which implies zero modality gap by definition; and (2) the MMD objective penalizes distributional discrepancies, further reducing any residual mismatch between modalities. We have verified this behavior across all datasets and will include the corresponding visualizations in the revised version. Due to space constraints, many of our extensive experiments were placed in the appendix; we will highlight key results, including the modality gap analysis, more prominently in the main paper.
>
> Finally, thank you for pointing out these minor issues. We will address them in the revised version.
>
> [1] Li et al. Csa: Data-efficient mapping of unimodal features to multimodal features. ICLR 2025.
>
> [2] Leeb et al. Partial alignment of representations via interventional consistency. ICLR 2025.
>
> [3] Wang et al. Omnibind: Large-scale omni multimodal representation via binding spaces. ICLR 2025.

---

> ### Comment · Reviewer_bSLV · 2025-08-07
> **Reply**
>
> I have raised my score to reflect the authors' strong rebuttal, which directly addressed several of my concerns by adding a several new empirical results and discussion. However, I believe fully integrating the additional discussion, clarifications, and results may benefit from a resubmission. Nevertheless, the work is very promising.

---

> > ### Author Response · Authors · 2025-08-08
> >
> > Thank you very much for your comment and support. We will include all new discussions and experiments made here in the final version of this manuscript.

---

### Note · Authors · 2025-08-13

Dear Reviewers and Area Chair,

We sincerely thank you for your time, thoughtful evaluations, and constructive feedback. We greatly appreciate the recognition of our paper’s contributions across multiple dimensions. In particular, we are encouraged by the acknowledgment of the **importance** of the problem we tackle (bSLV, MCZ3, bd36), the **originality** of our approach to multimodal learning with minimal paired data (bSLV), and the **clarity and motivation** underlying our methodology (bSLV, M2P3, bd36). We are also grateful for the appreciation of our paper’s **theoretical and empirical grounding** (bd36), as well as the positive remarks on our **extensive experiments** , **ablation studies** , and **visualizations**, all of which were noted as valuable in supporting and illustrating our method (bSLV, MCZ3, M2P3, bd36).

We were happy to see that our rebuttal adequately addressed the Reviewers' comments in detail, including additional experiments, clarifications of technical points, and expanded discussions, and led to all reviewers increasing their scores. Specifically, we appreciate the acknowledgement that: **(1)** The newly added baseline comparisons strengthened the empirical evaluation and further highlighted SUE’s advantage in the extreme low-pair regime (bSLV, M2P3, bd36). **(2)** Our clarifications on methodological assumptions, along with a theoretical justification addressed key conceptual questions (MCZ3). **(3)** Additional experiments on new modalities demonstrated SUE’s applicability beyond vision-language domains (MCZ3, bd36). **(4)** The new mAP results and scaling analysis reinforced the robustness of our findings (MCZ3). Following the reviewers' comments, we have incorporated all new results, clarifications, and discussions into the final version of the manuscript.

We are grateful for your constructive engagement, which has undoubtedly strengthened the paper, and we look forward to contributing this work to the community.

Thank you

---

### Decision · Program_Chairs · 2025-09-17

**Decision:**

Accept (poster)

**Comment:**

All reviewers suggest accepting the paper, so will be accepted. Please address the remaining concerns in the camera ready version.